# The bispecific innate cell engager AFM28 eliminates CD123+ leukemic stem and progenitor cells in AML and MDS

Nanni Schmitt [1,4], Jana-Julia Siegler[2,4], Alexandra Beck[1], Thomas Müller[2], Izabela Kozlowska[2], Séverine Sarlang[2], Uwe Reusch[2], Stefan Knackmuss[2], José Medina-Echeverz[2], Joachim Koch[2], Thorsten Ross[2], Ali Darwich [3], Lea Hoppe[1], Mohammed Abba[1], Alexander Streuer [1], Stefan Klein[1], Wolf-Karsten Hofmann[1], Anna Lisa Gündner[2], Christian Merz[2], Jan Endell[2], Jens Pahl [2,5] ✉ & Daniel Nowak [1,5] ✉

Strategies targeting leukemic stem and progenitor cells (LSPCs) are needed for durable remissions in acute myeloid leukemia (AML) and high-risk myelodysplastic neoplasms (MDS). While CD123 constitutes a promising target on LSPCs and leukemic blasts, previous CD123-targeting approaches showed limited efficacy and challenging safety profiles. Here, we describe the preclinical efficacy and safety of the bispecific CD123/CD16A innate cell engager "AFM28", demonstrating superior activity against AML and MDS patient-derived LSPCs and blasts in vitro compared to an Fc-enhanced CD123-targeting antibody, especially towards CD123low and/or CD64+ leukemic cells. AFM28 induces autologous anti-leukemic activity in fresh AML whole blood cultures, demonstrating its potential to enhance NK cell function from AML patients. Responsiveness can be further enhanced by allogeneic NK cell addition. Anti-leukemic activity of AFM28 is confirmed in xenograft mouse models. In addition, AFM28 is well tolerated and demonstrates pharmacodynamic activity in cynomolgus monkeys. Altogether, our results indicate that AFM28 has the potential to reduce relapse-inducing residual disease and promote long-term remissions for patients with AML and MDS with a favorable safety profile.

Acute myeloid leukemia (AML) and myelodysplastic neoplasms (MDS) are groups of heterogeneous myeloid neoplasms caused by clonal hematopoietic stem and progenitor cell disorders[1,2]. AML and high-risk (HR) MDS are characterized by blast infiltration of the bone marrow (BM) and differentiation arrest, causing ineffective hematopoiesis[3]. AML is one of the most common leukemias in adults, with approximately 20,000 new cases expected yearly in the US[4]. Despite available curative treatment options, AML has a poor prognosis, with a relative 5-year survival rate from diagnosis of approximately 30%[4]. The failure to induce long-term remission in AML has been linked to relapse-inducing residual disease driven by remaining therapy-resistant leukemic stem and progenitor cells (LSPCs)[5,6]. Accordingly, targeted elimination of leukemic blasts and LSPCs is critical to achieve long-term remission in AML and HR-MDS[5,7]. A promising therapeutic target is the interleukin-3 (IL-3) receptor subunit alpha (IL-3Rα/CD123), which is strongly expressed on blasts and LSPCs in AML and HR-MDS but is

[1]Department of Hematology and Oncology, Medical Faculty Mannheim, Heidelberg University, Mannheim, Germany. [2]Affimed GmbH, Gottlieb-Daimler-Straße 2, Mannheim, Germany. [3]Department of Orthopedic Surgery, Medical Faculty Mannheim, Heidelberg University, Mannheim, Germany. [4]These authors contributed equally: Nanni Schmitt, Jana-Julia Siegler. [5]These authors jointly supervised this work: Jens Pahl, Daniel Nowak. ✉e-mail: j.pahl@affimed.com; daniel.nowak@medma.uni-heidelberg.de

either absent or weakly expressed on healthy hematopoietic stem cells[8–10]. To this end, CD123 overexpression in AML was associated with poor prognosis and presence of residual disease[11,12]. Previous CD123-targeting approaches, such as Fc-enhanced IgG1-based antibodies and T cell engagers, including combinations with hypomethylating agents, showed unfavorable risk/benefit profiles in the clinic due to limited efficacy and/or challenging safety profiles, including the risk of cytokine release syndrome (CRS)[13–19]. Recent preclinical and early clinical anti-CD123 targeting approaches using novel CAR-T cell designs, antibody-drug conjugates, IL-3-diphtheria toxin fusion protein, as single agents or when combined with hypomethylating agents and/or the anti-BCL-2 inhibitor venetoclax, have suggested improved efficacy along with a manageable safety profile[20–24]. Moreover, the key mode of action of Fc-enhanced IgG1-based antibodies and NK cell engagers, antibody-dependent cellular cytotoxicity (ADCC), may have the potential to achieve meaningful clinical activity by leveraging the anti-leukemic activity of innate immune effector NK cells[20,25]. While NK cells show activity against leukemic blasts, their effect against LSPCs may be limited since they lack relevant natural ligands for NK cell receptors[26].

To maximize NK cell activity against leukemic cells, including LSPCs, the tetravalent bispecific CD123/CD16A innate cell engager AFM28 has been developed to enable high-affinity cross-linking of CD123+ leukemic cells with the NK cell activating receptor CD16A (FcγRIIIa) to mediate ADCC. This study demonstrates the preclinical efficacy and safety of AFM28, including the elimination of AML and HR-MDS patient-derived LSPCs in in vitro ADCC assays as well as ex vivo autologous and allogeneic whole blood cultures. In vivo activity of AFM28 is confirmed in AML mouse models and in cynomolgus monkeys, while being well tolerated and associated with a favorable cytokine release potential. Collectively, AFM28 demonstrates a promising efficacy and safety profile that has the potential to enable eradication of both leukemic blasts and LSPCs by leveraging endogenous innate immunity in some AML and HR-MDS patients or when being combined with allogeneic NK cell therapy.

## Results

### AFM28 induces potent and efficacious ADCC activity against AML cell lines irrespective of CD123 levels and mutational profile

By design, the tetravalent bispecific humanized IgG1-scFv fusion antibody AFM28 bivalently binds with high affinity in the picomolar range to human CD16A allelic variants and human CD123 with full cross-reactivity to cynomolgus CD16 and CD123 (see Supplementary Note 1, Supplementary Fig. 1A and Supplementary Table 1). To assess AFM28-mediated cytotoxicity in the context of the expected genetic and phenotypic heterogeneity of AML, ADCC assays were performed using leukemia cell lines with variable CD123 expression levels and mutational profiles. AFM28 showed high ADCC potency and efficacy irrespective of the levels of CD123 expression (Fig. 1A, B) and mutational profiles, including mutations conferring poor prognosis (e.g., TP53) (Supplementary Table 2). AFM28 did not induce target cell lysis against CD123− OPM-2 cells; no cell lysis occurred in the presence of a non-targeting RSV/CD16A control engager devoid of CD123 targeting (Fig. 1A, B). Furthermore, there was no lysis of bystander CD123− CD30+ Karpas-299 cells in the presence of CD123+ EOL-1 cells and AFM28, while Karpas-299 cells were otherwise susceptible to ADCC by a CD30-targeting engager (Supplementary Fig. 1B). Collectively, these results indicate the specificity of AFM28-induced ADCC towards CD123+ cells. AFM28-induced ADCC was associated with increased NK cell degranulation and IFN-γ expression, and the up-regulation of the activation markers CD137 and CD25 on NK cells in response to CD123+ target cells (Supplementary Fig. 1C, D).

AFM28 enhanced the cytotoxicity of NK cells with higher efficacy and potency compared with an Fc-enhanced anti-CD123 IgG1 control antibody, in particular towards cell lines with lower CD123 expression levels (Fig. 1A, B). The Fcγ receptors CD64 (FcγRI) and CD32 (FcγRII)

are frequently overexpressed on tumors, including AML cell lines and can impair the efficacy of therapeutic IgG1 antibodies via cis or trans interaction with their Fc portion[27,28]. Therefore, we assessed whether AFM28-induced ADCC was impacted by FcγR expression on CD123+ leukemic cells. The Fc-enhanced anti-CD123 IgG1 did not mediate ADCC against the CD64high AML cell lines SKM-1, OCI-AML3 and THP-1 (Fig. 1A, C, D). In contrast, AFM28 induced ADCC irrespective of CD64 or CD32 expression, indicating that AFM28 retained ADCC efficacy against CD123+ AML cell lines co-expressing CD64 and CD32.

### AFM28 enables depletion of primary CD123+ blasts from AML and HR-MDS patients in combination with allogeneic NK cells

To assess the ADCC efficacy of AFM28 against diverse AML and HR-MDS BM samples, allogeneic NK cells from healthy donors were co-cultured with primary BMMC samples from AML (n = 10) and HR-MDS (n = 5) patients containing primary CD34+ CD38+ or CD33+ CD38+ leukemic cells, hereafter referred to as blasts, for 24 h (Supplementary Fig. 2A–C). AFM28 specifically induced a significant, concentration-dependent reduction of CD123+ blasts (Fig. 2A–D), resulting in 79% (range 50–100%) and 70% (range 37–86%) lysis of the CD123+ blast population in AML and HR-MDS, respectively. Of note, AFM28-mediated ADCC was highly effective even when blasts displayed very low CD123 expression levels (Supplementary Fig. 3A). In addition, on the remaining AML blasts, the expression of inhibitory molecules like PD-L1 and HLA-I was explored after an extended 48 h co-culture. While the anti-leukemic activity was maintained, exposure to allogeneic NK cells led to an increase in PD-L1 expression on the remaining blasts, albeit to a very heterogeneous extent between the AML samples, with a trend for a further increase in the presence of AFM28; minor effects were observed for HLA-I (Supplementary Fig. 4A–C). Finally, when comparing the efficacy of AFM28 to that of the Fc-enhanced anti-CD123 IgG1, there was a significant trend for superior AML blast reduction by AFM28 in comparison to the Fc-enhanced anti-CD123 IgG1, in particular towards leukemic blasts with high CD64 expression (Fig. 2E–H and Supplementary Fig. 2E).

### AFM28 eradicates primary CD123+ LSPCs in AML and HR-MDS patient samples and spares healthy hematopoiesis

Next, we assessed the ability of AFM28 to induce ADCC by allogeneic healthy NK cells against LSPCs that were phenotypically defined as CD34+ CD38− CD117+ cell population (means of 0.3% and 6.8% in patient AML and HR-MDS BMMCs, respectively) (Fig. 3A and Supplementary Fig. 2A–D). AFM28 induced a significant reduction of CD123+ LSPCs, resulting in 98% (range 95–99%) depletion of the LSPC population in both AML (n = 5) and HR-MDS (n = 5) (Fig. 3B, C).

To substantiate these findings, the colony-forming potential of AML (n = 5), HR-MDS (n = 5) and healthy (n = 5) BMMC-derived CD34+ cells after treatment with AFM28 and allogeneic NK cells was assessed. AFM28 induced a reduction of colony numbers of 62% (range 35–94%), 58% (range 24–84%) and 25% (range 20–40%) for AML, HR-MDS and healthy samples, respectively, compared with CD34+ cells without exposure to AFM28 and/or NK cells (Fig. 3D–G). In summary, these data demonstrate that AFM28 can mediate NK cell-dependent reduction of primary LSPCs of AML and HR-MDS patients while largely sparing healthy progenitors.

### AFM28 induces killing of CD123+ blasts in primary AML samples by patient-derived NK cells

Since at least CD16A-independent NK cell cytotoxicity is often impaired in AML patients due to disease-related cellular defects[29–31], we subsequently investigated whether AFM28 can induce anti-leukemic activity by AML patients' residual endogenous innate immunity in cultures of freshly-drawn peripheral blood (PB) samples from newly diagnosed AML patients (n = 6), suggestive for the single-agent efficacy potential of AFM28 in AML patients. Remarkably, AFM28

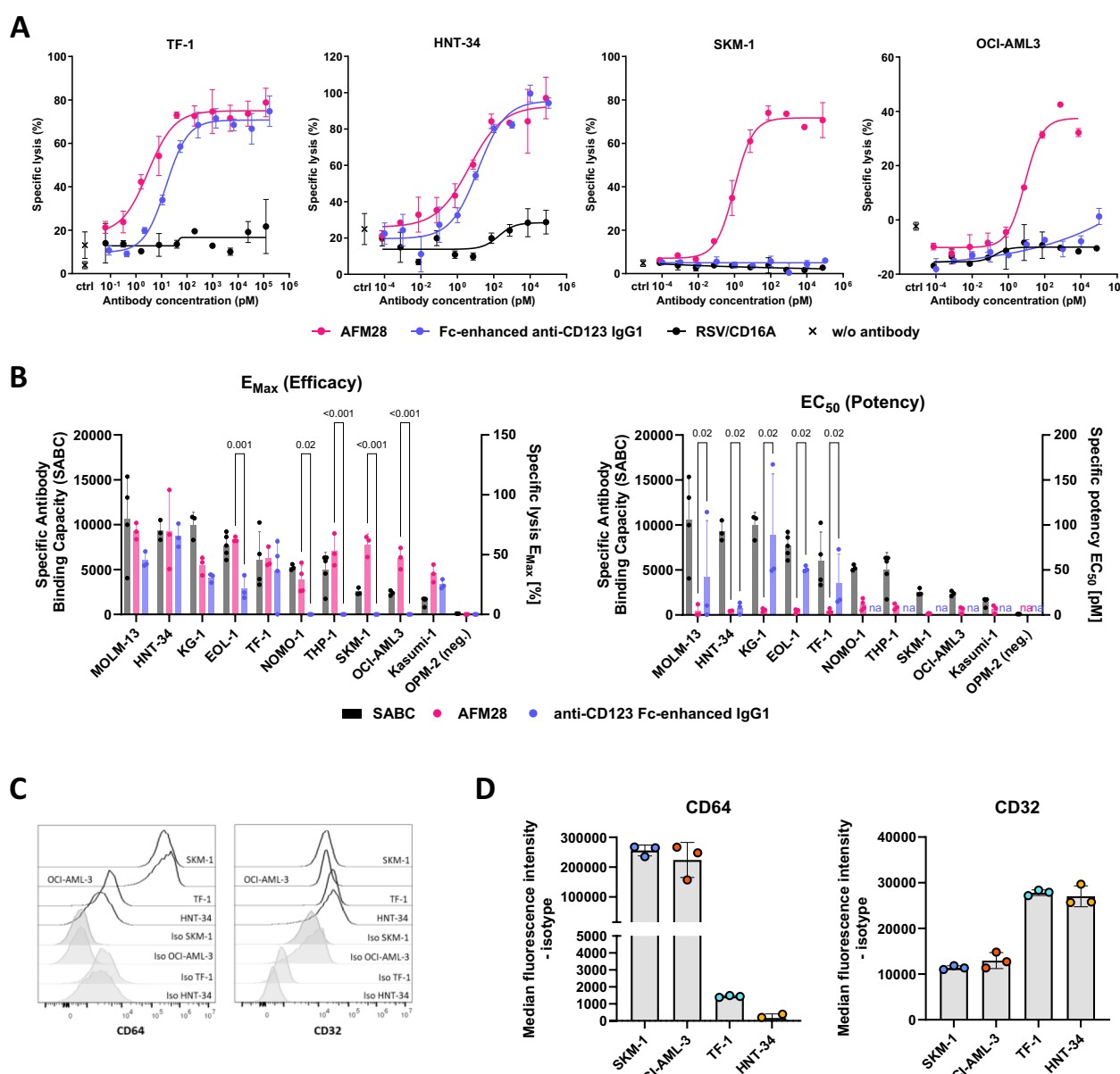

**Fig. 1 | AFM28 induces lysis of CD123$^+$ leukemic cell lines irrespective of CD123 and CD64 expression levels.** Buffy coat-derived allogeneic healthy donor NK cells were cultured at a 2:1 E:T ratio with calcein-labeled leukemic cell lines in the presence of increasing concentrations of AFM28, Fc-enhanced anti-CD123 IgG1 or non-targeting control RSV/CD16A. **A** Exemplary data for selected tumor cell lines. Specific tumor cell lysis of indicated CD123$^+$ leukemic cells by NK cells derived from one healthy donor was quantified by calcein-release cytotoxicity assay. Data depict one biological sample represented as the mean ± SD of two technical replicates. **B** Gray bars: Quantification of CD123 expression levels on the cell surface of leukemic cell lines determined as SABC measured by flow cytometry, represented as mean ± SD of 3, 4 or 5 independent biological replicate experiments, as indicated by individual data points per cell line. Magenta and blue bars: Efficacy ($E_{max}$) and potency ($EC_{50}$) of tumor cell lysis by NK cells induced by AFM28 (magenta bars) and

anti-CD123 Fc-enhanced IgG1 (blue bars) for all tested cell lines was quantified by calcein-release cytotoxicity assay represented as mean ± SD of 3 or 4 independent biological replicate experiments, as indicated by individual data points per cell line. Note that each experiment utilized NK cells of an independent healthy donor source. Data were analyzed using two-way ANOVA and Šídák's multiple comparisons test, with $p$-values below 0.05 indicated in the figure. na, not applicable since EC50 values were not reached. **C** Representative histogram of CD64 and CD32 expression of the indicated leukemic cell lines. **D** Quantitative analysis of the MFI of CD64 and CD32 on indicated cell lines (MFI of isotype control antibodies subtracted), represented as mean ± SD of three independent experiments ($n = 3$). MFI median fluorescence intensity; SABC specific antibody binding capacity; SD standard deviation.

demonstrated effective reduction of blasts in the PB of 50% of patients ($n = 3/6$) without addition of allogeneic NK cells (Fig. 4A). Responders showed 76% lysis (range 59–90%), while non-responders exhibited 7% lysis of blasts (range 0–22%). Both responders and non-responders showed heterogeneous frequencies of endogenous NK cells, ranging from 0.5:1, 0.1:1, and 0.1:1 effector-to-target (E:T) ratios for responders to 0.01:1, 0.1:1 and 0.2:1 E:T ratios for non-responders, with low and

high CD16A expression (Supplementary Fig. 3D). Furthermore, it was tested whether the anti-leukemic activity could be augmented by the addition of allogeneic, healthy NK cells, using fresh, non-expanded NK cells ($n = 4$) and one standardized preparation of cryopreserved, cytokine-expanded NK cells (see Supplementary Methods and Supplementary Fig. 7). On average, basal anti-leukemic activity of all NK cells was similar (Fig. 4A), likely attributed to the freezing/thawing

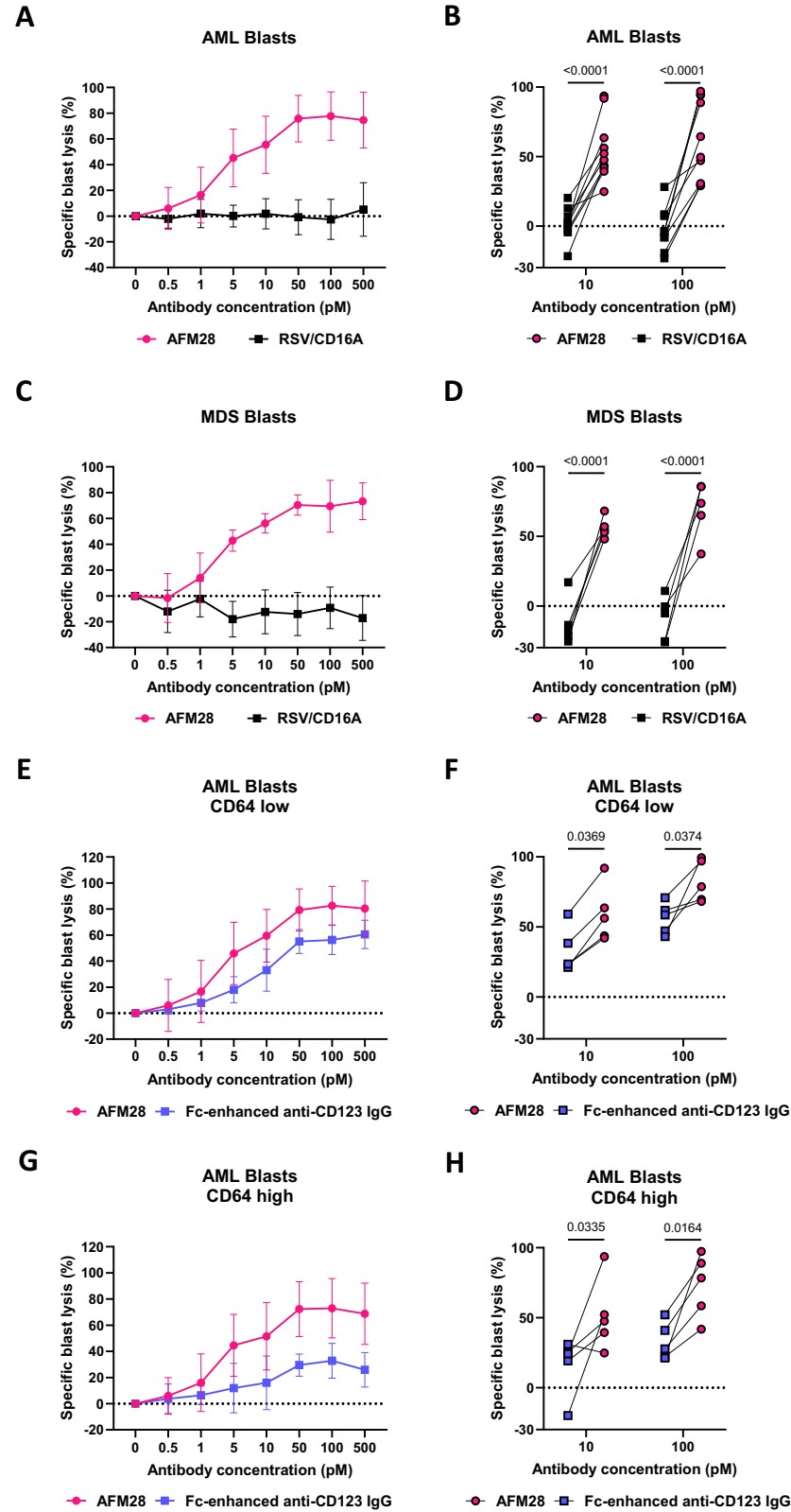

procedure of the healthy cytokine-expanded NK cells[32,33]. Overall, the combination of AFM28 and allogeneic NK cells enhanced the anti-leukemic activity, increasing the proportion of responders (lysis > 50%) to 100% ($n = 6/6$) and overall increasing blast lysis to 87% (range 60–100%).

To further assess the cytotoxic activity of AML patient-derived NK cells in combination with AFM28, NK cells were enriched from AML patients ($n = 5$) and tested for their ADCC potential on BMMCs from a single AML patient, compared to healthy NK ($n = 5$) cells. Effective lysis of allogeneic blasts was achieved by $n = 3$ AML-derived and all $n = 5$ healthy NK cell samples (Fig. 4B). Collectively, these results suggest that AFM28 can effectively arm AML patients' endogenous NK cells for anti-leukemic activity to a comparable extent to healthy donors in a subgroup of AML patients.

**Fig. 2 | AFM28 efficiently directs allogeneic NK cells to CD123⁺ blasts in AML and HR-MDS patient BMMC samples.** BMMC samples derived from AML and MDS patients were treated with 0–500 pM of AFM28, a non-targeting control (RSV/CD16A) or an Fc-enhanced anti-CD123 IgG antibody for 24 h in the presence of IL-2-preincubated allogeneic healthy donor NK cells, all derived from different donors, at a 1:1 E:T ratio. Analysis was performed using flow cytometry. Blasts were defined as viable/CD45[low]/CD34⁺ or CD33⁺/CD38⁺/CD123⁺ cells. The gating strategy is shown in Supplementary Fig. 2A. Cell counts of 0 pM treatment were set to baseline.

**A–D** Concentration-dependent lysis of blasts from AML (**A**, **B**, n = 10) or MDS (**C**, **D**, n = 5) patients by AFM28 compared to RSV/CD16A. **E–H** Concentration-dependent lysis of blasts from AML patients with low (**E**, **F**, n = 5) or high (**G**, **H**, n = 5) CD64 MFI by AFM28 compared to an Fc-enhanced anti-CD123 IgG antibody. Data in (**A**, **C**, **E** and **G**) are represented as mean ± SD. Data in (**B**, **D**, **F** and **H**) were analyzed using two-way ANOVA and Šídák's multiple comparisons test. MFI median fluorescence intensity; SD standard deviation.

## AFM28 blocks IL-3Rα signaling

IL-3 induces proliferation and survival of leukemic cells through the alpha subunit of IL-3R (CD123) and downstream STAT5 phosphorylation[34,35]. Hence, the ability of AFM28 to specifically inhibit IL-3 signaling in the absence of NK cells was evaluated against the CD123⁺ TF-1 cell line, which requires IL-3 or GM-CSF for cell growth[36]. AFM28 specifically inhibited IL-3-dependent TF-1 cell growth, whereas GM-CSF-dependent cell growth remained unaffected (Fig. 5A, B). Consequently, AFM28 abrogated STAT5 phosphorylation in IL-3–stimulated TF-1 cells, whereas STAT6 phosphorylation, induced by IL-4[37], was not affected (Fig. 5C, D), corroborating specificity of AFM28 to the IL-3–IL-3Rα–STAT5 signaling axis.

## AFM28 induces anti-leukemic activity against AML in vivo

Next, we investigated whether the in vitro activity of AFM28 translates into anti-leukemic activity in vivo using a disseminated AML cell line model in mice transgenic for human CD16A expressed on endogenous murine innate immune effector cells. For that purpose, irradiated CB17.SCID hCD16A mice received intravenous (IV) infusion of the human AML cell line EOL-1_Luc and were subjected to increasing doses of AFM28, ranging from 0.3, 1.0 and 3 mg/kg, or vehicle treatment followed by monitoring of tumor burden by total body bioluminescence imaging (Fig. 6A). Importantly, AFM28 administration reduced tumor growth compared to vehicle-treated littermates in a dose-dependent manner (Fig. 6B), resulting in significantly prolonged survival versus vehicle control. The median increased life span relative to vehicle control (median survival [mS]: 38 days) was 66% (+ 25 days mS) for the 3 mg/kg dose, 50% (+ 19 days mS) for the 1 mg/kg dose and 34% (+ 13 days mS) for the 0.3 mg/kg dose at the experimental cutoff of 70 days after tumor inoculation (Fig. 6C). Moreover, in vivo anti-leukemic control by the combination of AFM28 with allogeneic human NK cells was tested in irradiated fully immunodeficient hIL-15 NOG mice inoculated IV with EOL-1_Luc cells (Fig. 6D). Tumor growth was markedly delayed when mice were treated with AFM28-armed NK cells, whereas AFM28 treatment alone failed to induce tumor control in this model due to the lack of responsive effector cells (Fig. 6E). Adoptive transfer of NK cells alone showed limited anti-leukemic activity and only at early time points in this model.

## AFM28 shows a favorable cytokine release profile and pharmacodynamic activity in human whole blood and cynomolgus monkeys

The potential of AFM28 to stimulate ADCC-induced release of proinflammatory cytokines and chemokines was determined in leukocyte cultures and in an ex vivo whole blood assay system (see Supplementary Methods), both containing CD16A⁺ effector cells (e.g., NK cells) and CD123⁺ target cells (e.g., basophils)[38]. In leukocyte cultures, there was a modest increase in the release of cytokines such as IFN-γ, IL-6, TNF, and GM-CSF after exposure to AFM28 (Fig. 7A). In contrast, a CD123/CD3 T cell engager, used as a positive control, demonstrated a much stronger (10- to 100-fold) induction of cytokine release, notably inducing more than 1000 pg/mL of the proinflammatory, CRS-associated cytokines IFN-γ, IL-6, and TNF[39]. In circulating human whole blood cultures derived from healthy donors, AFM28 exposure induced low levels of cytokine release (Supplementary Fig. 5A).

Nevertheless, distinct pharmacodynamic (PD) activity was observed in the leukocyte cultures and circulating human whole blood cultures in form of a time-dependent and concentration-dependent decline in CD123⁺ basophils and CD123⁺ plasmacytoid dendritic cells (pDC) (Fig. 7B, C and Supplementary Fig. 5B). AFM28 PD activity was accompanied by increased CD137, CD25 and CD69 expression on NK cells (Fig. 7B, C and Supplementary Fig. 5B), which is consistent with the activation of NK cells as a result of ADCC (Supplementary Fig. 1D). While Fc-mediated targeting of CD16A by IgG1 antibodies is associated with shedding of CD16A[40], in this study, there was a trend for an increase in CD16A expression on NK cells in the presence of AFM28 in whole blood cultures and in response to CD123⁺ target cells (Fig. 7C and Supplementary Fig. 6A). In this context, it was noted that NK cell exposure to AFM28 could inhibit PMA/ionomycin-induced CD16A shedding while PMA/ionomycin-induced NK cell activation markers CD69 and CD137 were not affected (Supplementary Fig. 6B). An Fc-enhanced anti-CD123 IgG1 antibody did not inhibit PMA/ionomycin-induced CD16A shedding. These results suggest that Fc-independent bivalent high-affinity targeting of CD16A by AFM28 can interfere with activation-induced shedding of CD16A and maintain CD16A expression upon ADCC.

The tolerability of AFM28 and PD activity was tested in cynomolgus monkeys (n = 36) representing the relevant toxicology animal model for AFM28 due to AFM28's species cross-reactivity to cynomolgus CD16 and cynomolgus CD123. Animals received weekly IV AFM28 doses of 4, 20 or 100 mg/kg, as well as a vehicle control, over a 4-week period and a 4-week recovery phase. Overall, no adverse clinical observations were noted. PD activity was demonstrated by the expected depletion of circulating CD123⁺ cells, i.e., basophils and pDC, as indicated by the rapid and dose-dependent reduction in absolute HLA-DR⁻ FcεR1a⁺ basophil counts and HLA-DR⁺ BDCA-2⁺ pDC counts at all doses within 4 h of the first dose that was sustained for two to four weeks (Fig. 7D and Supplementary Fig. 5C). In contrast, vehicle-treated control animals only showed a transient drop after the first vehicle infusion, which returned to baseline within one week. Toxicokinetic analysis (non-compartmental) demonstrated that animals were systemically exposed to AFM28 levels, observing dose linearity in the first week. However, several low- and mid-dose animals revealed reduced exposure due to an emerging anti-drug antibody (ADA) response from day 15 onwards, potentially leading to the clearing and/or neutralizing activity of anti-AFM28 antibodies emerging after this period. In this context, the complete depletion of basophils was observed for only two weeks, followed by a slow recovery of cell counts, which thus coincided with the emerging ADA response (Fig. 7D). Monitoring cytokine and chemokine analysis revealed no signs of CRS upon 2 h IV infusion of AFM28. Minor, transient release of circulating IL-6 was measurable after the first AFM28 dose (Fig. 7E). In some animals, substantial serum IL-6 levels were detected after infusion at later doses that also correlated with the emerging ADA response, suggesting the formation of immune complexes resulting in modest induction of systemic IL-6. It should be noted that the emergence of the ADA response in cynomolgus monkeys, potentially a xenogeneic effect, is not predictive for ADA formation in humans[41]. Overall, repeated doses of up to 100 mg/kg/week of AFM28 were well tolerated without evidence for CRS risk factors.

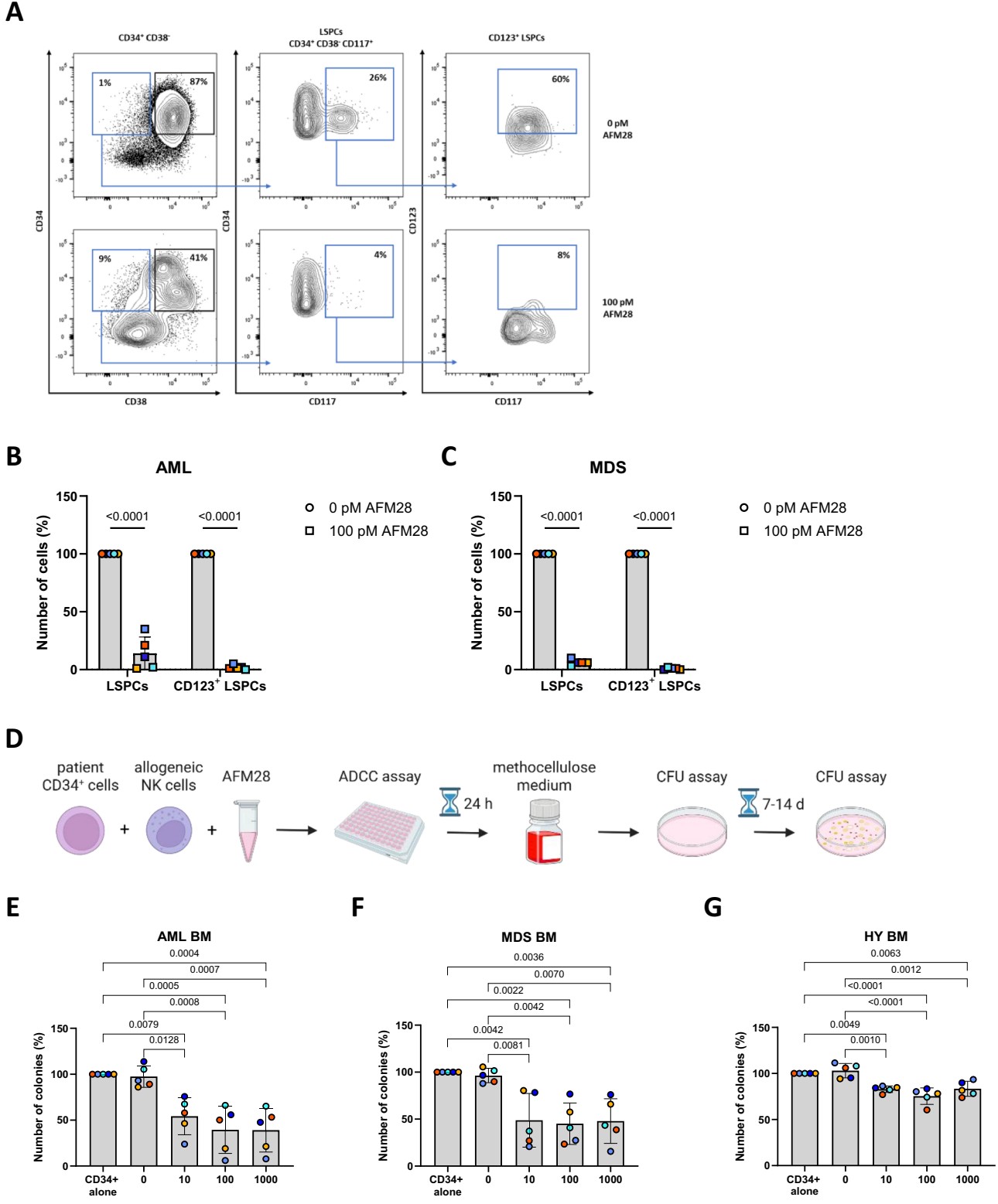

## Discussion

Relapse and refractoriness after prior treatments caused by residual disease driven by therapy-resistant LSPCs confer a poor prognosis for patients with AML and HR-MDS[5,7]. The bispecific NK cell engager, AFM28, was designed to leverage the anti-leukemic potential of NK cells towards CD123+ malignant cells, including LSPCs. In the present study, AFM28 demonstrated to specifically bind to CD16A (FcγRIIIA) and CD123, mediating the depletion of AML cell lines via ADCC irrespective of their levels of CD123, CD64 expression or mutational profile (e.g., TP53). Concomitantly, AFM28-mediated ADCC effectively reduced leukemic blasts and LSPCs from BM samples of patients with AML and HR-MDS. In an autologous ex vivo setting, AFM28 induced anti-leukemic activity in fresh AML whole blood cultures, demonstrating AFM28's potential to effectively arm AML patients' NK cells. Responsiveness was enhanced by the addition of allogeneic NK cells and could convert non-responders to responders. Overall, AFM28

**Fig. 3 | AFM28 induces potent lysis of CD123⁺ LSPCs in AML and HR-MDS patient samples and spares healthy hematopoiesis.** BMMC patient-derived AML and MDS samples were treated with or without 100 pM of AFM28 for 24 h in the presence of IL-2-preincubated allogeneic NK cells, all derived from different donors, at an E:T ratio of 1:1. Analysis was performed using flow cytometry. LSPCs were defined as viable/CD45ˡᵒʷ/CD34⁺/CD38⁻/CD117⁺ cells. The gating strategy is shown in Supplementary Fig. 2A. Cell counts of 0 pM treatment for LSPCs and CD123⁺ LSPCs were set to baseline. **A** Representative dot plot of LSPC lysis following treatment with 100 pM AFM28 or without AFM28 (indicated as 0 pM) of one AML patient sample. **B, C** Lysis of LSPCs from AML patients (**B**, *n* = 5) and from MDS patients (**C**, *n* = 5) in the presence of 100 pM AFM28 or without AFM28. Data are represented as mean ± SD and were analyzed using one-way and two-way ANOVA and Šídák's multiple comparisons test. **D**, Schematic workflow of the CFU assay. Created in BioRender (https://BioRender.com/8l2brjy). **E–G** CFU assay results of AML (**E**, *n* = 5), MDS (**F**, *n* = 5) and healthy (**G**, *n* = 5) CD34⁺ cell samples treated with 0/10/100/1000 pM of AFM28 for 24 h in the presence of allogeneic NK cells at an E:T ratio of 1:1. "CD34⁺ only" describes culturing untreated CD34⁺ cells without allogeneic NK cells. Colonies were counted manually. Colony count of "CD34⁺ only" condition was normalized to 100%. Data are represented as mean ± SD and were analyzed using one-way ANOVA and Tukey's multiple comparisons test. SD standard deviation.

**A**

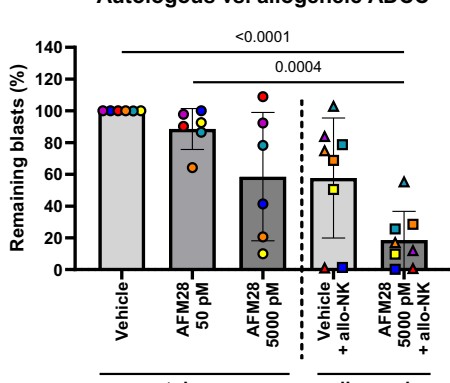

**B**

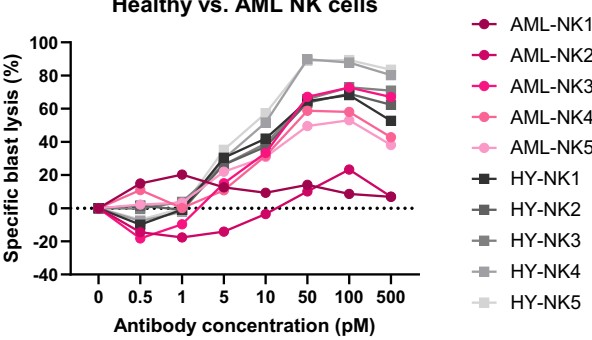

**Fig. 4 | AFM28 enables autologous and allogeneic ADCC of patient-derived NK cells against CD123⁺ blasts in primary AML samples.** **A** Fresh whole blood samples from newly diagnosed AML patients (*n* = 6) were treated with AFM28 (0/0.01/1 μg/ml equivalent to 0/50/5000 pM) or vehicle in the absence (circles) or presence of healthy donor-derived allogeneic NK cells (squares: fresh, non-expanded NK cells; triangles: cryopreserved cytokine-expanded NK cells generated by a standardized protocol, see Supplementary Methods and Supplementary Fig. 7), all derived from different donors, for 24 h at an E:T ratio (NK cells:peripheral leukocytes) of 1:1. Blasts were defined as viable/CD45ˡᵒʷ/CD123⁺/CD33⁺/CD117⁺ or CD34⁺/CD117⁺. The gating strategy is shown in Supplementary Fig. 2A. Leukemic blast counts of vehicle treatment without NK cells were set to baseline. E:T ratios of patient's endogenous NK cells to blasts (in the absence of healthy donor-derived allogeneic NK cells) were 0.1:1, 0.01:1 and 0.2:1 for non-responders and 0.5:1, 0.1:1 and 0.1:1 for responders (data points from top to bottom for 5000 pM AFM28). Data are represented as mean ± SD and were analyzed using one-way ANOVA and Tukey's multiple comparisons test. **B** BMMC samples derived from a single AML patient were treated with 0–500 pM of AFM28 for 24 h in the presence of allogeneic AML patient-derived NK cells (*n* = 5) or healthy donor-derived NK cells (*n* = 5) at a 1:1 E:T ratio. Analysis was performed using flow cytometry. Blasts were defined as viable/CD45ˡᵒʷ/CD34⁺/CD38⁻/CD123⁺. Blast counts of 0 pM treatment were set to baseline. HY healthy, SD standard deviation.

achieved maximal ADCC activity in vitro at concentrations of approximately 100 pM. AFM28 reduced tumor growth and prolonged survival in a xenogeneic model of disseminated AML. Importantly, systemic application of AFM28 into cynomolgus monkeys was well-tolerated, without any signs of cytokine release syndrome, while demonstrating PD activity. Taken together, our data suggest that AFM28 may offer an effective treatment option due to its favorable safety profile and pharmacological activity in relevant in vitro and in vivo models. While AFM28 monotherapy has the potential to mediate anti-leukemic activity by leveraging the endogenous innate immunity in some AML patients, a combination of AFM28 with allogeneic NK cell immunotherapy may be an option to improve responses in patients with low numbers and/or hypofunctional NK cells.

While ADCC is a promising mechanism to leverage the full anti-leukemic potential of NK cells, conventional IgG1-based anti-CD123 approaches remained behind expectations in the clinic[13,14]. Beyond other factors, limited efficacy may be attributed to the expression of CD64 (FcγRI) on AML cells, which is the high-affinity FcγR receptor for IgG1 antibodies and has been shown to interfere with ADCC of IgG1 antibodies[42,43]. In accordance with previous studies, we confirm that high CD64 expression on AML cell lines and primary leukemic blasts from AML patients abolished ADCC by an Fc-enhanced anti-CD123 IgG1 antibody[20,27,44]. In contrast, we show that AFM28 does not bind to CD64, resulting in substantially increased ADCC efficacy of AFM28 against all tested CD64⁺ cell lines. In this context, AFM28-mediated ADCC activity was also evident against AML cell lines and leukemic blasts with very low CD123 target antigen expression levels (below 200 antigens per cell as approximated by CD123-specific antibody binding capacity (SABC)). Therefore, AFM28 is expected to exert broad therapeutic activity in patients with variable or low CD123 expression levels and high CD64 expression. The observed activity in AML cells carrying adverse prognostic lesions such as TP53 suggests that this therapy may have potential in combination with current therapies, which may be less effective in the context of such lesions. It remains to be explored in future preclinical studies whether baseline or therapy-induced PD-L1 expression, for instance by ADCC-driven IFN-γ[45], affects the activity of cellular immunotherapies in AML, while on the other hand providing opportunities for potential combinations of AFM28 with PD-1 negative NK cell products or PD-1/PD-L1 checkpoint therapy.

Relapse in AML is attributed to a rare, chemotherapy-resistant cellular subpopulation of leukemic stem cells[5,6]. A high proportion of leukemic stem cells at the time of diagnosis is predictive for residual disease, suggesting their significant contribution to disease relapse[46,47]. In addition, more committed leukemic progenitor cells have been indicated to enable disease propagation[5,48]. Therefore, novel therapies that are able to eradicate both leukemic stem and progenitor cells are warranted. In our study, AFM28-mediated depletion of LSPCs was assessed using a phenotypic marker combination (CD34⁺ CD38⁻ CD117/c-kit⁺) that has been reported to detect the majority of LSPCs characterized by their repopulation activity in serial xenotransplantation experiments and long-term culture-initiating

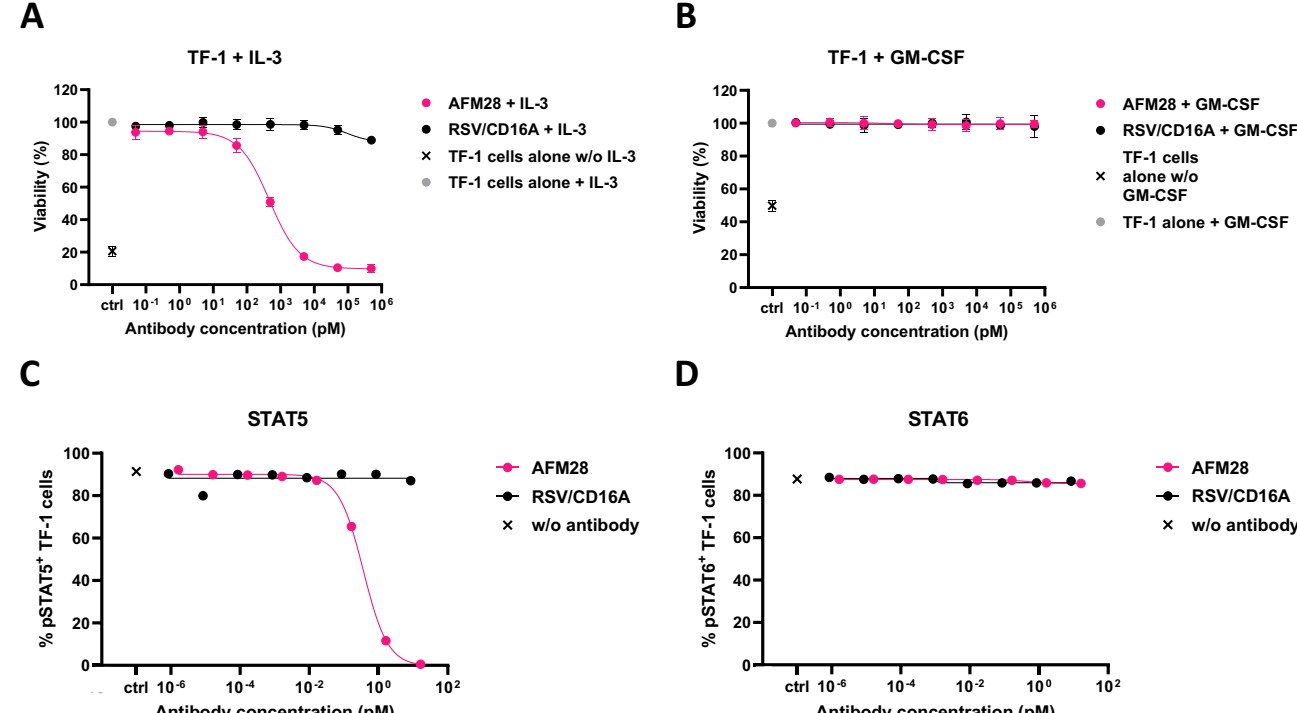

**Fig. 5 | AFM28 blocks IL-3R signaling inhibiting STAT5 phosphorylation in TF-1.**
**A**, **B** TF-1 cell growth was stimulated with IL-3 (**A**) or GM-CSF (**B**) in the presence of titrated AFM28, a non-targeting control (RSV/CD16A), or without antibody addition and with IL-3/GM-CSF (gray dot) or without antibody addition and without IL-3/GM-CSF (cross), for 72 h at 37 °C. The relative number of metabolically active viable cells was determined using the CellTiter-Glo assay in three independent experiments. The results shown were normalized to the maximum cell viability signal, which represents 100%. Data shown as mean ± SD (*n* = 3). **C**, **D** Intracellular phosphorylated STAT5 or STAT6 was measured in TF-1 cells upon stimulation with IL-3 (**C**) or IL-4 (**D**) in the presence of titrated AFM28, a non-targeting control engager (RSV/CD16A) or without antibodies (cross). One representative experiment is shown out of three performed. SD standard deviation.

activity in vitro[5,6,47]. In addition, we substantiated this approach by assessing the colony formation potential of total BMMC-derived CD34+ cells[6,49]. The results of our two approaches indicate that AFM28 can induce the depletion of CD123+ LSPCs and blasts via NK cells in CD34+ samples of patients with AML and HR-MDS, resulting in fewer formed colonies, corroborating that LSPCs were indeed reduced. Notably, colony formation of healthy CD34+ samples was only modestly affected, suggesting that AFM28 may have less myelosuppressive activity against healthy hematopoietic progenitors as compared to anti-CD123 CAR-T cells[50]. Previously, it has been suggested that leukemic stem cells can escape NK cell recognition due to the reduction of NKG2D ligands[26]. Consequently, AFM28 enables LSPC recognition by NK cells and, therefore, may counteract evasion of LSPCs from NK cell immunosurveillance. Overall, the capacity of AFM28 to trigger eradication of LSPCs via CD123 holds promise for eliminating these key drivers of disease relapse, thus suggesting the potential to improve depth of response and long-term remissions. It remains to be investigated whether additional targeting of activating and inhibitory NK cell ligands expressed by leukemic blasts, including LSC-like cells[5] enhances ADCC to further restrict the risk for MRD.

Previous evidence has also shown that IL-3 may stimulate AML blast proliferation[11]. In this study, AFM28 could directly inhibit tumor cell metabolic activity via blockade of IL-3-induced STAT5 phosphorylation. Accordingly, AFM28 exhibits an additional mechanism of action that is independent of CD16A+ immune effector cell-mediated killing. The therapeutic contribution of this mechanism is not fully elucidated, as prior approaches focusing on IL-3 inhibition did not show meaningful efficacy[13].

A key feature of innate cell engagers is their potent anti-neoplastic efficacy in vivo[51,52]. CB17.SCID mice are a versatile tool to evaluate the therapeutic anti-leukemic efficacy of AFM28, since these mice harbor murine endogenous innate immune cells like NK cells with functional human CD16A[53,54]. Strong tumor growth inhibition as well as a significant increase in survival was observed for AFM28 at all dose levels tested, demonstrating the biological activity of AFM28 against human AML in vivo. In addition, by using fully immunodeficient mice, we could demonstrate the proof of concept for anti-leukemic control via the combination of AFM28 with allogeneic human NK cell immunotherapy.

While novel drugs need to exhibit a promising activity profile, patient safety is a key factor for their application, contributing to risk-benefit considerations. In circulating human whole blood cultures as well as leukocyte cultures used in this study, AFM28 led to the depletion of CD123+ target cells; however, the release of proinflammatory cytokines was substantially lower for AFM28 than for a CD123/CD3 T cell engager. Therefore, AFM28 may harbor low risk for CRS or other immune-related toxicities, which remain a challenge for a broad, safe- and convenient application of T cell-based approaches[13]. Moreover, in cynomolgus monkeys, the potential of AFM28 to induce on-target PD activity in vivo was confirmed by depletion of CD123+ basophils and CD123+ pDC, while no adverse side effects were detected. IL-6 release was only observed at later time points associated with emerging ADA, thus suggesting a potential relationship to respective immune complexes. The relevance for the human application remains unclear, as the immunogenicity of AFM28 with its human framework would be expected to be lower in humans compared to cynomolgus. Taken together, provided that the preclinical safety data translate into the clinic, the NK cell-directed mode of action of AFM28 may offer higher specificity and lower toxicity compared with conventional chemotherapeutics or (CAR-)T-cell based approaches, which entail a greater risk for CRS, endothelial injury and hematopoietic toxicities[47,55].

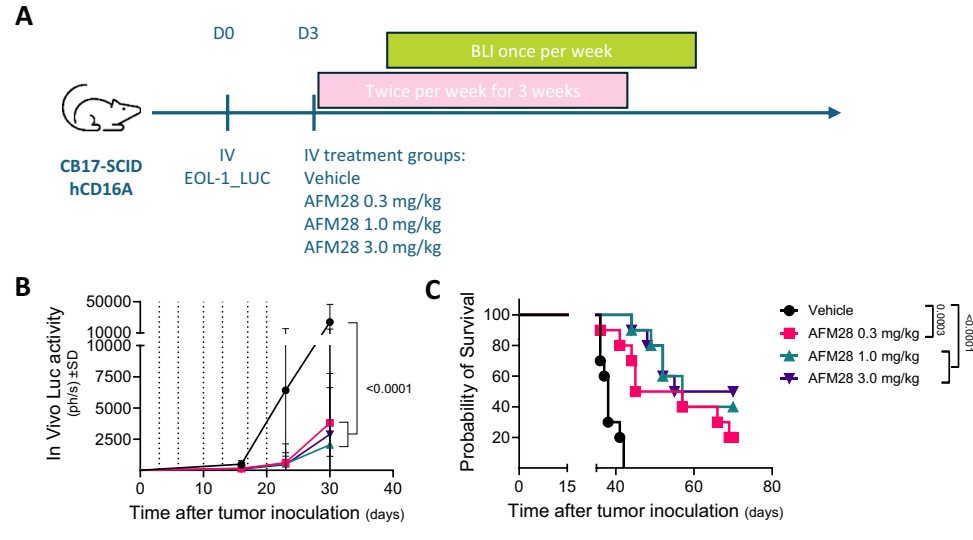

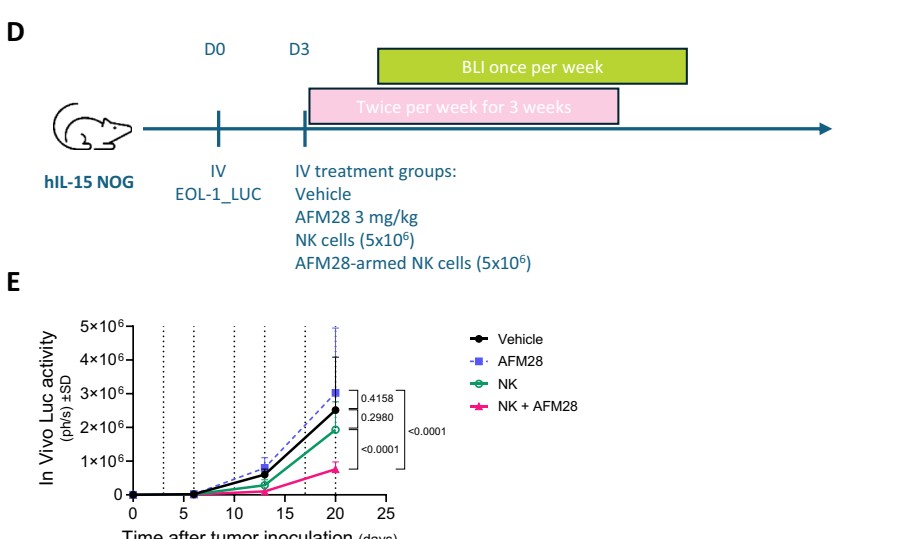

**Fig. 6 | In vivo anti-leukemic activity of AFM28 against a human AML cell line.**
**A** Experimental schedule of CB17.SCID hCD16A xenograft model. On day 0,
CB17.SCID hCD16A mice ($n = 40$ mice, 10 mice/group) received 1.2 Gy irradiation,
followed by IV injection of $1 \times 10^6$ EOL-1_Luc cells four hours later. Three days after
tumor cell inoculation, mice were randomized according to body weight (10 mice/
group) and received either vehicle or AFM28 at different doses (0.3/1/3 mg/kg)
twice per week for a total of 3 weeks. BLI measurements were performed every
week starting day 15. Dotted lines represent treatment days. **B** Tumor growth
represented by in vivo luciferase expression upon treatment is shown as photons
per second in a linear scale. Data are represented as mean ± SD ($n = 40$ mice, 10
mice/group). Significance was tested by two-way ANOVA and Turkey's multiple
comparisons test comparing treatment groups at day 30, with $p$-values indicated in
the figure. **C** Kaplan-Meier plot shows survival upon treatment. Significance
between groups was tested by the Log-rank (Mantel-Cox) test, with $p$-values indi-
cated in the figure. **D** Experimental schedule of hIL-15 NOG xenograft model. On day

0, hIL-15 NOG mice ($n = 32$ mice, 8 mice/group) received 1.2 Gy irradiation, followed
by IV injection of $1 \times 10^6$ EOL-1_Luc cells four hours later. Three days after tumor cell
inoculation, mice were randomized according to body weight (8 mice/group) and
received vehicle, AFM28 (3 mg/kg), NK cells ($5 \times 10^6$) or AFM28-armed NK cells
($5 \times 10^6$) twice per week for a total of 3 weeks. Cryopreserved cytokine-expanded
NK cells were generated by a standardized protocol (see Supplementary Methods
and Supplementary Fig. 7). BLI measurements were performed every week starting
day 6. Dotted lines represent treatment days. **E** Tumor growth represented by
in vivo luciferase expression upon treatment is shown as photons per second in a
linear scale. Data are represented as mean ± SD ($n = 32$ mice, 8 mice/group). Sig-
nificance was tested by two-way ANOVA and Turkey's multiple comparisons test
comparing treatment groups at day 20, with $p$-values indicated in the figure. BIW
twice per week; BLI bioluminescence imaging; D day; hCD16A human CD16A; IV
intravenous; Luc luciferase.

The clinical efficacy of AFM28 is anticipated to be dependent on
the functionality and frequency of NK cells in patients. NK cells from
patients with AML have been reported to exert CD16A-mediated ADCC
activity[29,30], demonstrating that CD16A-mediated NK cell cytotoxicity
in AML patients is mostly conserved[20,29,30]. Despite this, it has been
reported that at least the natural CD16A-independent cytotoxic activ-
ity of NK cells can be lower in patients with AML compared to the
natural activity observed in individuals without AML[29,31,56]. Aside from

immunosuppression, this may be attributed to reduced expression
and function of naturally cytotoxic receptors such as NKp30 and
NKp46[29,30]. Hence, leveraging CD16A-mediated ADCC holds the pro-
mise to maximize the anti-leukemic potential of NK cells of AML
patients. NK cells that develop post-chemotherapy/stem cell trans-
plant have been shown to have an immature phenotype[57,58]; despite
this, they can induce comparable lysis of AML blasts in the presence of
an anti-CD123 IgG1 antibody[59]. Overall, retained ADCC activity of

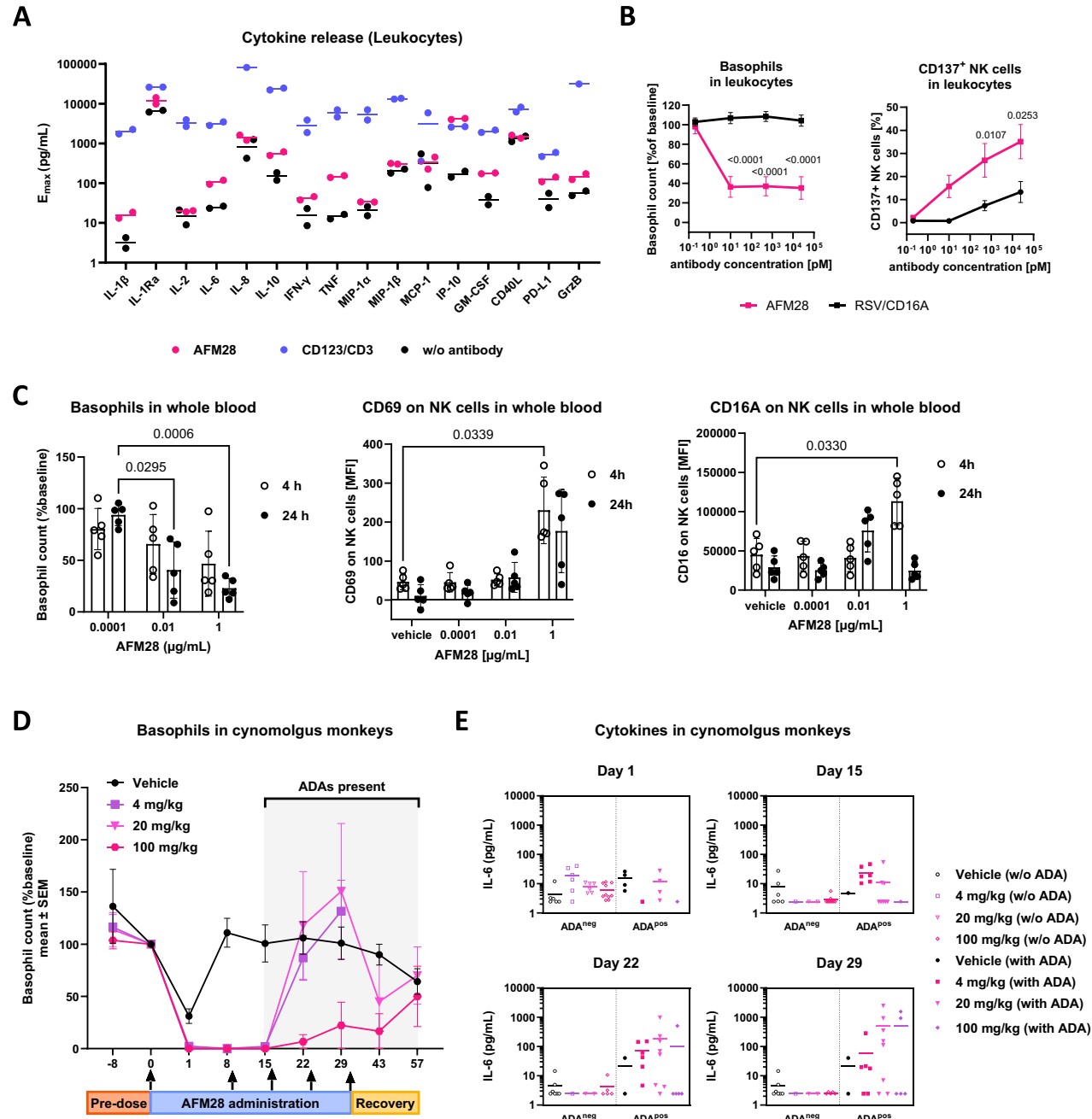

**Fig. 7 | AFM28 induces basophil depletion and shows a lower cytokine release than a T cell-engager. A** Comparison of cytokine release from primary human leukocytes treated with AFM28, a CD123/CD3 T cell engager or without antibody (w/o) after 24 h incubation. $E_{max}$ denotes maximal induction of the respective cytokine. Cytokines were measured using the Luminex 24-plex human cytokine panel (only the 16 induced cytokines are shown). Individual data from two donors are shown. **B** Dose-dependent depletion of CD123+ basophils and dose-dependent changes in CD137 expression on NK cells after 24 h incubation of primary human leukocytes in the presence of titrated AFM28 or RSV/CD16A control engager (for further details and gating strategy see Supplementary Fig. 5B). Data are represented as mean ± SD of six donors and were analyzed using two-way ANOVA and Šídák's multiple comparisons test. **C** Circulating healthy donor whole blood loops in the presence of titrated AFM28. Dose-dependent depletion of CD123+ peripheral blood basophils and dose-dependent changes in CD69 and CD16A expression on NK cells. The anti-CD16 3G8 antibody does not compete with AFM28 for CD16A binding, as further illustrated in Supplementary Fig. 6 and as previously shown for the innate cell engager acimtamig (AFM13) targeting the same CD16A epitope[66]. For gating

strategy see Supplementary Fig. 8. The E:T cell ratio between NK cells and basophils ranged between 2.1-9.2 amongst donors and did not correlate with the extend of basophil depletion. Individual data from $n = 5$ donors are shown (% remaining basophils, normalized versus formulation buffer). The population was normally distributed tested by the Shapiro-Wilk test, and significance was tested by two-way ANOVA and Turkey's multiple comparisons test. Data are represented as mean ± SD. **D** AFM28 induces basophil depletion in cynomolgus monkeys, indicating PD activity towards CD123+ cells (gated as CD3−/CD14−/CD20−/CD159a−/HLA−DR−/ FcεR1a+). Basophil recovery after day 15 was concurrent with ADA presence and corresponding loss of exposure. Arrows indicate dosing occasions. Data are represented as mean ± SD ($n = 36$ animals; 10 animals per vehicle, 20 mg/kg and 100 mg/kg groups; 6 animals per 4 mg/kg group). **E** Increases in IL-6 levels upon treatment of cynomolgus monkeys ($n = 36$ animals; 10 animals per vehicle, 20 mg/ kg and 100 mg/kg groups; 6 animals per 4 mg/kg group) with AFM28 were detected only at later dosing time points correlating with the onset of an ADA response, mostly at 4 and 20 mg/kg doses of AFM28. ADA anti-drug antibodies.

patient-derived NK cells has been demonstrated in several other tumor indications, underscoring its immunotherapeutic potential[60–62]. To enable therapeutic activity, the number of CD16A⁺ NK cells in patients needs to be sufficiently high in addition to retained ADCC functionality; the latter may indeed be the case in the majority of patients with AML reported to be positive for CD16A and responsive to CD16A engagement at diagnosis and after chemotherapy[29–31]. In line with that, we show in our ex vivo autologous fresh AML whole blood cultures and additionally using freshly enriched NK cells from AML patients at diagnosis that AFM28 can enable the anti-leukemic activity of endogenous NK cells at least in some responders, suggesting the potential of AFM28 monotherapy activity in such patients. However, if selected patients lack sufficient numbers of fully functional NK cells, which may be the case for AML patients after relapse and even after achieving complete remission[63,64], our data suggest that this could be addressed by combining AFM28 with an allogeneic NK cell therapy to possibly enable conversion of non-responders to responders. The promising potential of such an approach has been demonstrated recently for the bispecific CD30/CD16A innate cell engager acimtamig (AFM13), when combined with cord blood-derived NK cells, resulting in preclinical and clinical efficacy with an overall response rate of 94% and a complete response rate of 71% at the recommended phase 2 dose in heavily pretreated patients with Hodgkin lymphoma and non-Hodgkin lymphoma[48,65].

In conclusion, AFM28 is a CD123-targeting innate cell engager that induced highly potent and selective lysis of CD123⁺ LSPCs via NK cells. By eliminating LSPCs that are a key player in residual disease, AFM28 treatment might enable more durable remission in AML and MDS patients. Based on the efficacious pharmacological activity and the positive preclinical safety profile, AFM28 may overcome limitations of previous CD123-targeting approaches. To investigate this further, a clinical phase 1 dose escalation study is currently recruiting to assess the safety and tolerability of AFM28 monotherapy in patients with relapsed or refractory AML (NCT05817058).

## Methods

### Primary AML and MDS patient samples
Primary BM and PB samples from AML ($n = 15$ BM; $n = 11$ PB) and HR-MDS ($n = 7$ BM) patients were obtained from residual diagnostic BM aspirations and blood draws. Healthy BM ($n = 5$) was isolated from femoral heads received after hip replacement surgery. All samples were collected after obtaining the patients' written informed consent and in accordance with the Declaration of Helsinki. The sampling and use of all human material has been approved by the Institutional Review Board "Ethikkommission II" of the Medical Faculty Mannheim, Heidelberg University, Germany. Details on the BM and PB donor characteristics and for which assays the samples were used are provided in Supplementary Tables 3–5.

### Cell lines
The cell lines MOLM-13 (ACC-554, RRID:CVCL_2119), KG-1 (ACC-14, RRID:CVCL_0374), EOL-1 (ACC 386, RRID:CVCL_0258), THP-1 (ACC-16, RRID:CVCL_0006), Kasumi-1 (ACC 220, RRID:CVCL_0589), OPM-2 (ACC-50, RRID:CVCL_1625), and NOMO-1 (ACC-542, RRID:CVCL_1609) were cultured in RPMI-1640 medium (Invitrogen, cat. no. 21875-091) supplemented with 10% FCS (Life Technologies, cat. no. 10438026 (lot 2286117RP) and 1% glutamine (Invitrogen, cat.no. 25030-024). HNT-34 (ACC 600, RRID:CVCL_2071) was cultured in RPMI-1640 medium supplemented with 10% FCS. TF-1 (ACC-334, RRID:CVCL_0559) was cultured in RPMI-1640 medium supplemented with 20% FCS and 2 ng/mL granulocyte macrophage colony-stimulating factor (GM-CSF, PeproTech, cat. no. 300-03). SKM-1 (ACC-547, RRID:CVCL_0098) was cultured in RPMI-1640 medium supplemented with 20% FCS. OCI AML3 (ACC-582, RRID:CVCL_1844) was cultured in alpha-MEM

(Invitrogen, cat. no. 22571-020), supplemented with 20% FCS and 1% glutamine. EOL-1 cells overexpressing Luciferase (EOL-1 Luc) were generated at Experimental Pharmacology & Oncology Berlin-Buch GmbH. All cells were cultured in the presence of 1% penicillin/streptomycin (P/S) (Invitrogen, cat. no. 15140-122) and were purchased from DSMZ.

### Cell isolations
BMMCs, PBMCs and leukocytes were isolated from buffy coats of healthy donors (commercially obtained from the Red Cross Germany, Mannheim) or from BM and PB of AML and HR-MDS patients by density gradient centrifugation and/or red cell lysis, respectively. Cells were used directly or maintained in RPMI-1640 medium supplemented with 10% FCS, 2 mM L-glutamine, 1% P/S (referred to as complete RPMI medium) for a maximum of 24 h until use. NK cells were enriched from PBMCs using the EasySep™ Human NK Cell Enrichment kit (STEMCELL Technologies, cat. no. 19055) or the NK Cell Isolation kit (Miltenyi Biotec, cat. no. 130-092-657) according to the manufacturer's instructions. If indicated, freshly isolated NK cells were cultured overnight in RPMI-1640 supplemented with 10% FCS, 1% P/S and 100 U/mL IL-2 (STEMCELL Technologies, cat. no. 78036.3) and used for experiments the following day. The purity of NK cell isolation was determined by flow cytometry, typically demonstrating > 80% CD56⁺ cells (data not shown). Enrichment of CD34-positive (CD34⁺) cells from BMMCs was performed using the CD34 MicroBead kit (Miltenyi Biotec, cat no. 130-046-702), typically demonstrating > 90% CD34⁺ cells (data not shown). BMMCs and CD34⁺ cells were viably frozen until further use.

### Depletion of primary cells in BM samples
Prior to all assays, non-expanded and expanded NK cells were intracellularly stained with 0.5 μM CellTracker Green CMFDA Dye (Thermo Fisher, cat. no. C7025) at a cell density of $10^6$ cells/mL. For specific blast lysis analysis, $2.5 \times 10^5$ non-expanded allogeneic NK cells derived from AML patients or healthy donors were co-cultured with $2.5 \times 10^5$ primary BMMCs from AML and MDS (allogeneic setup only) patients in the presence of CD123/CD16A AFM28, an Fc-enhanced anti-CD123 IgG1 antibody (allogeneic setup only) or a non-targeting RSV/CD16A control (allogeneic setup only) (0/0.5/1/5/10/50/100/500 pM) in triplicate. For LSPC lysis analysis, $2 \times 10^6$ NK cells were co-cultured with $2 \times 10^6$ primary BMMCs in singlicate with or without AFM28 (100 pM). For autologous ex vivo ADCC assays, freshly drawn PB of AML patients was treated with AFM28 (0/50/5000 pM) or vehicle in the absence or presence of healthy allogeneic NK cells under modest stirring to prevent sedimentation of cells. As indicated, allogeneic NK cells represented either fresh, non-expanded NK cells derived from four healthy donors, or one cryopreserved, cytokine-expanded NK cell preparation derived from one healthy donor using a standardized protocol (see Supplementary Methods), used immediately after thawing. All assays were incubated for 24 h. Afterwards, cells were blocked with human Fc receptor blocking reagent (Miltenyi Biotec, cat. no. 30-059-901) and subjected to antibody staining (Supplementary Table 6) to identify blasts and LSPCs. In case of autologous ex vivo ADCC assays, PB was lysed before staining. Dead cells were excluded using SYTOX Blue (Thermo Fisher, cat. no. S34857). The gating strategy and antibodies used are shown in Supplementary Fig. 2A and Supplementary Table 6, respectively.

### Colony-forming unit (CFU) assay
Allogeneic non-expanded CMFDA-labeled NK cells and primary AML, HR-MDS and healthy CD34⁺ cells mixed at an E:T ratio of 1:1 were treated with AFM28 at different concentrations (0/10/100/1000 pM) and incubated for 24 h. As a control, CD34⁺ cells were cultured without NK cells. After 24 h, cells were mixed with semi-solid MethoCult H4435 Enriched medium (STEMCELL Technologies, cat. no. 04435) and

plated in multiple replicates. After incubation for 7–14 days, colonies were counted manually.

## Calcein-release ADCC assay

Calcein-labeled target cells (10 mM, Life Technologies, cat. no. C3100MP) were seeded at $1 \times 10^4$ cells per well of a U-bottom 96-well microplate in 100 μL of complete RPMI medium and supplemented with NK cells at the indicated E:T ratios (or complete RPMI only as a control) in a final volume of 200 μL without or with indicated concentrations of antibody constructs (or complete RPMI only as a control). After 4 h incubation at 37 °C at 5% $CO_2$, the release of fluorescent calcein (F) into cell-free supernatants was quantified as a measure for target cell death at 520 nm using a plate reader (EnSight, Perkin Elmer). The specific cell lysis was calculated by the following formula: [F(sample)−F(spontaneous)]/[F(maximum)−F(spontaneous)] x 100%. F(spontaneous) represents the fluorescence from target cells cultured in the absence of effector cells and antibodies. F(maximum) represents the total cell lysis induced by addition of 1% TritonX100 (Roth, cat. no. 3051.2). Mean values of specific target cell lysis (%) and standard deviations (SD) were plotted; EC50 potency values were determined by fitting the nonlinear regression model to sigmoidal dose-response curves (variable slope) using Prism (v10, GraphPad Software).

## IL-3-dependent viability of CD123+ TF-1 cells

TF-1 cells were seeded in triplicate at $2.5 \times 10^4$ cells/well in 96-well plates. TF-1 cell growth was stimulated with 0.5 ng/mL IL-3 (PeproTech, cat. no. 200-03) or 2 ng/mL GM-CSF (PeproTech, cat. no. 300-03). Cell viability, inferred by the relative number of metabolically active cells, was assessed after 72 h incubation at 37 °C in the presence of titrated AFM28, or a non-targeting RSV/CD16A control, or without antibody using the CellTiter-Glo® assay (Promega, cat. no. G7572).

## Cytokine release from human leukocytes and basophil depletion in whole blood loops (WBL)

In vitro cytokine release assays using human healthy donor leukocytes ($2 \times 10^6$ leukocytes/well) were seeded into U-bottom 96-well microtiter plates with increasing concentrations of AFM28 (starting at $5 \times 10^5$ pM followed by eight 10-fold dilutions) in a total volume of 200 μL/well. In addition, a CD123/CD3 T cell engager with known induction of clinical cytokine release syndrome was used as a positive control antibody. Plates were incubated for 24 h at 37 °C and 5% $CO_2$ in a humidified incubator before centrifugation at $70 \times g$ for 2 min at room temperature. Cell culture supernatants (80 μL) were harvested from each well, and cytokines were quantified by bead-based multiplex methodology (Human Immunotherapy Magnetic Luminex Performance Assay 24-plex Fixed Panel, R&D Systems). Furthermore, a multi-parametric ex vivo assessment of AFM28 in human healthy donor fresh blood employed the ID. Flow WBL system (Immuneed AB, Uppsala, Sweden) was performed. Briefly, AFM28 (0.5, 50 and 5000 pM) or a buffer control (w/o) was added to freshly-drawn whole blood samples and kept in circulation for up to 24 h. Assessment of hematology (cell counts) was performed using Sysmax, release of 36 cytokines and chemokines was analyzed using the MSD human 36-plex array, and depletion of basophils, gated as CRTH2+, CD123+, CD3- (Supplementary Fig. 8), was carried out at the indicated time points.

## Flow cytometry

For extracellular staining of surface proteins, $0.2–1 \times 10^6$ cells were incubated with 100 μL of mouse anti-human antibodies (Supplementary Table 6) alongside Fixable Viability Dye eFluor 780 (Thermo Fisher, cat. no. 65-0865-14) in PBS (Invitrogen, cat. no. 14190-169) containing 2% heat-inactivated FCS (Invitrogen, cat. no. A31608-01) and 0.1% sodium azide (Roth, cat. no. K305.1). Intracellular staining of

IFN-γ was performed using the Cyto-Fast™ Fix/Perm Buffer Set (Bio-Legend). CD123 target antigen expression on cells was quantified via the CD123-SABC using 10 μg/mL anti-CD123 mAb clone 7G3, followed by F(ab´)2 fragment of FITC-conjugate goat anti-mouse IgG using the QIFIKIT (Agilent Dako). For intracellular staining of phosphorylated signal transducer and activator of transcription 5 (STAT5) and STAT6, TF-1 cells were starved overnight in 1% FCS in RPMI-1640 medium, then stimulated with 10 ng/mL IL-3 or 5 ng/mL IL-4 (Sigma-Aldrich, cat. no.: SRP4137) in RPMI-1640 medium supplemented with 10% FCS for 30 min at 37 °C. The assay was performed in the presence or absence of titrated AFM28 or non-targeting RSV/CD16A, followed by fixation and cell lysis (Fixation buffer, BioLegend, cat. no. 420801; True-Phos Perm Buffer, BioLegend, cat. no. 425401). Analyses were performed using BD FACSCelesta (Becton Dickinson), CytoFlex or CytoFlexS (Beckman Colter) flow cytometers and the BD FACSDiva software (version 9.0.1) or CytExpert (version 2.4.0.28, Beckman Colter), respectively. Data were analyzed using FlowJo software (FlowJo LLC) and CytExpert (Beckman Colter).

## In vivo anti-leukemic activity

To assess the in vivo anti-leukemic activity of AFM28 by endogenous effector cells, forty 18-week-old female CB17.SCID hCD16A mice (CB17.SCID-Fcgr4tm2(FCGR3A)/Bcgen, Biocytogen, cat. no. 111174) were irradiated with 1.2 Gy to facilitate tumor cell engraftment, followed by IV injection of $1 \times 10^6$ EOL-1_Luc cells 4 h later. On day 3, mice were block randomized into groups (10 mice/group) based on body weight. Mice were treated either with vehicle or increasing doses of AFM28 (0.3/1/3 mg/kg) twice per week for a total of 3 weeks (10 mice/group). Systemic tumor growth was assessed by bioluminescence imaging (BLI) once per week starting on day 15 after tumor inoculation. To assess in vivo tumor activity of AFM28 in combination with adoptive transfer of human NK cells, thirty-two 18-week-old female hIL-15 NOG mice (NOD.Cg-Prkdc$^{scid}$ Il2rg$^{tm1Sug}$ Tg(CMV-IL2/IL15)1-1Jic/JicTac, Taconic Biosciences, cat. no. 13683-F) were irradiated and injected IV with $1 \times 10^6$ EOL-1_Luc cells. On day 3, mice were block randomized into groups (8 mice/group) based on body weight, followed by treatment with vehicle, AFM28 alone (3 mg/kg), non-armed allogeneic NK cells and AFM28-armed allogeneic NK cells twice per week for a total of 3 weeks. The allogeneic NK cells had been expanded from one healthy donor using a standardized protocol (see Supplementary Methods and Supplementary Fig. 7) and were tested immediately after thawing at one dose of $5 \times 10^6$ cells per injection per mouse. Systemic tumor growth was assessed by BLI once per week starting on day 6 after tumor inoculation. All mouse studies were conducted at Experimental Pharmacology & Oncology (EPO) Berlin-Buch GmbH. All mice were housed in a specific pathogen-free environment and were handled, fed, bred, and maintained in the animal facilities at EPO according to institutional guidelines. Mice were anesthetized once a week with isoflurane and received intraperitoneally 150 mg/kg D-Luciferin dissolved in PBS. BLI measurements were performed under anesthesia with isoflurane with the NightOwl II LB983 in vivo imaging system. IndiGO 2.0.5.0 software was used for initial BLI analysis, color-coding of the signal intensity and quantification. Body weights and general health conditions were monitored daily according to the Guidelines for the Welfare and Use of Animals in Cancer Research. Body condition scores were also used to monitor the development of systemic lymphoma burden daily. Information from preliminary pilot studies were used to define the maximal permitted lymphoma burden and adopted in the body condition scores. The maximal tumor size/burden was not exceeded during the studies. Animals were individually euthanized, under anesthesia by cervical dislocation, if any of the ethical endpoints were reached, i.e., decline in health status, body weight loss, and body condition score loss. All animals were housed under the following conditions: room temperature, 24 ± 2 °C; relative humidity 50 ± 10 %; light period artificial, 12 h dark/12 h light rhythm. The study designs

were reviewed and approved by the Landesamt für Gesundheit und Soziales (LAGeSo), Berlin, Germany.

## Repeat-dose toxicology study in cynomolgus monkeys

AFM28 doses (0/4/20/100 mg/kg) were administered IV to cynomolgus monkeys once per week for 4 weeks (five doses in total; 3–5 animals/sex/group), followed by a 4-week recovery phase. AFM28 safety was assessed based on clinical observations, body weights, food consumption, ophthalmology, clinical and anatomic pathology, along with integrated safety pharmacology endpoints including cardiovascular and respiratory assessments. Histopathology evaluation was performed on a standard tissue panel. Further endpoints included toxicokinetics, immunogenicity, and cytokine release. As a PD endpoint, circulating CD123+ basophil granulocytes (CD3−/CD14−/CD20−/CD159a−/HLA-DR−/FcεR1a+) and CD123+ pDC (CD3−/CD14−/CD20−/CD159a−/HLA-DR+/BDCA-2+) were quantified. Absolute cell counts, that had been computed from relative percentages determined by flow cytometry and absolute leukocyte counts determined by manual leukocyte differential of blood smears, were normalized to day 0 and depicted as a percentage of baseline. The study was conducted by Labcorp Early Development Services GmbH in Münster (Germany) and was approved by Landesamt für Natur, Umwelt und Verbraucherschutz Nordrhein-Westfalen (LANUV) ethical committee (details in Supplementary Methods).

## Statistical analysis

Individual group data were analyzed using descriptive analysis (mean [standard deviation], median [interquartile range]). Statistical analysis was performed using Prism (v10, GraphPad Software). Either one- or two-way analysis of variance (ANOVA) with multiple comparisons test were used to determine statistical significance between treatment groups. For tumor-bearing mice survival after treatment, increased life span (%) was calculated using the following formula: Increased life span (%) = $\frac{\text{median survival treatment} - \text{median survival control}}{\text{median survival control}} \times 100$. A $p$-value of < 0.05 was considered statistically significant.

## Reporting summary

Further information on research design is available in the Nature Portfolio Reporting Summary linked to this article.

# Data availability

The data generated in this study are provided in the Supplementary Information/Source Data file. Unique reagents are available from the authors. Source data are provided in this paper.

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

## Acknowledgements

We thank Verena Nowak, Julia Obländer, Melda Göl and Iris Palme for excellent technical assistance. We thank Joshua Alcaniz, Maria Stecklum and Jens Hoffmann (Experimental Pharmacology & Oncology Berlin-Buch GmbH) for their advice and troubleshooting regarding conduction of mouse studies. Editorial assistance was provided by Meridian Healthcomms Ltd., funded by Affimed.

## Author contributions

N.S., J.J.S., J.P., D.N., and C.M. conceived and designed the study. D.N., J.E., C.M., J.K., and T.R. supervised the study. Experiments were designed and performed by N.S., J.J.S., J.P., A.B., T.M., I.K., S.S., U.R., T.R., J.M.E., and SKnackmuss. W.K.H. provided research infrastructure. A.D., L.H., M.A., A.S., and SKlein provided patient material and clinical data. Data were analyzed and interpreted by N.S., J.J.S., J.P., A.B., T.M., I.K., J.M.E., A.L.G., and S.S. All authors were involved in the writing or review of the manuscript and approved the final version.

## Funding

## Competing interests

This study was supported by funding from Affimed GmbH. J.P. is an employee of Affimed GmbH. J.J.S., S.S., U.R., J.M.E., J.K., A.L.G., T.R., T.M., I.K., SKnackmuss, C.M., and J.E. were employees of Affimed at the time of writing and execution of this study. D.N. has received research funding from Affimed, Tolero Pharma, Pharmaxis and Apogenix, received funding for a clinical trial from AbbVie and received honoraria from Celgene and Takeda. The remaining authors declare no competing interests.
