## [Transparent Peer Review file · Nature Communications]

The bispecific innate cell engager AFM28 eliminates CD123+ leukemic stem and progenitor cells in AML and MDS

Corresponding Author: Professor Daniel Nowak

Version 0:

Reviewer comments:

Reviewer #1

(Remarks to the Author)

This is another paper on a bispecific antibody in the setting of AML. The difference to most of the prior reports, is that here an innate bispecific is introduced, that is recruiting NK cells through CD16 engagement. Preclinical data, from in vitro to in vivo are shown. The data demonstrates efficacy against primary AML and MDS cells and demonstrates efficacy in vivo (mouse model) and safety in a cynomolgus monkeys.

The preclinical in vitro and in vivo data set is sort of "standard" for the regulatory affairs, but for a publication in this journal is missing aspects that are of biological relevance and interest to the scientific community. With the current data set, this is just another paper demonstrating efficacy preclinical, but is not answering any questions on existing or evolving resistance and is not tackling questions of patient derived NK cells or impact of the bone marrow environment on the CD123-CD16a bispecific construct activity. Therefore I like the authors to share some more data on the following aspects:

1. The authors demonstrate the secretion of proinflammatory cytokines by CD123-CD16 activated NK cells - can you please show the impact on the phenotype of NK cells (CD16 downregulation has been reported), but importantly also the impact on AML cells ? e.g. upregulation of HLA class I molecules (inhibitory to NK cells), upregulation of PD-L1 ? The authors need to extend the time of coculture to pick up immune escape mechanisms. How do residual AML cells look after coculture ? is there an outgrowth after removal of the antibody ? At least the impact on the cells within the coculture should be addressed.
2. Also, all the experiments were done in an "allo" setting with NK cells from healthy donors. Hence, it would be very interesting to look at i) NK cells from AML patients derived from different time points, e.g. initial diagnosis vs relapse but also ii) in cocultures with complete autologous set up , or just add the construct to primary bone marrow samples.
3. The E:T ratio used is very unphysiological - I wonder if mimicking the E:T ratio in a patient, and demonstrating NK cell expansion over time might better reflect the clinical situation.
4. The authors need to integrate already running and reported clinical trails with NK-cell recruiting bispecifics as well as CART activities. This is missing in the discussion and also in the introduction, including Fc optimized anti-CD123 trials, which were applied in conjunction with AZA (to upregulate CD123 and increase NK cell activity), but failed to show any benefit.
5. After CD123 CART therapy severe bone marrow aplasia has been observed. Please extend the data set on safety, also considering the fact that IFN γ has been reported to upregulate CD123 on endothelial cells. Hence the monkey model, without any target cells, and hence no secretion of proinflammatory cytokines is not the suitable animal model.

Reviewer #2

(Remarks to the Author)

Please see the attachment.

Reviewer #3

(Remarks to the Author)

This is a generally well written report on the pre-clinical development of the CD123 innate cell engager AFM28. The experiments are generally well designed and the methods and figures clearly delineated. As acknowledged by the authors CD123 is an attractive target for leukaemia-directed therapy with a view to directly eliminating LSC populations and/or interrupting IL3 trophic signalling. Despite its attractiveness, effective clinical targeting of CD123 with various agents has proven elusive. In the current paper Siegler and colleagues has described the next logical step in the development of CD123-directed therapy with the development of an NK-engaging tetra-valent molecule. The experiments that are outlined in the paper are a logical sequence of pre-clinical work up of a new drug but overall do not reveal any particularly new findings into the therapeutic application of this ICE. The sensitivity of AML cells to allogeneic NK cells in the presence of AFM28 is a reasonable experimental design, but would have been more clinically relevant if performed using NK cells autologous to the AML targets. Presumably the samples containing leukaemic targets would also contain some residual NK populations. The confounding effect of HLA-disparity/non-self recognition of leukaemic targets by allogeneic NK cells limits the interpretation of this finding. Importantly, the authors discuss the role of NK cell fitness in AML patients and its potentially limiting effect in ICE-based therapies. Whilst there is generally (sufficiently) preserved NK cell numbers and function in patients with newly diagnosed AML, further consideration and discussion should be undertaken regarding how this may change following chemotherapy delivery and/or in the setting of relapsed disease, which is likely to reflect the population in which these novel agents are to be initially trialed. Further consideration and discussion should be undertaken as to the expression of inhibitory NK receptor ligands on LSCs targets, which may limit NK cell activation in vivo.

Version 1:

Reviewer comments:

Reviewer #1

(Remarks to the Author)

Thank you for the revision and integrating data of additional experiments. I have 3 questions / comments to the presented novel data sets:

1. While CD69 is frequently used as an early activation marker for NK cells, it has significant limitations in terms of specificity and functional correlation. CD69 is an early and transient marker that does not necessarily correlate with functional cytotoxicity or sustained NK cell activation. This makes it a less reliable indicator of specific NK cell-target interactions unlike markers such as CD107a (degranulation) or IFN- γ production
2. Looking at the cytokine profile it is hard to believe that IFN γ and TNF do not compromise activity, as there is a high amount secreted and it is known and published, that these cytokines lead to upregulation of inhibitory checkpoint molecules. Most likely the assay time is too short to assess this adaptive resistance factor, however in the patient this is likely to occur. Can you look at the AML blasts, e.g. in a less favorable E:T ratio and check for upregulation after e.g. 72 hours ?
3. Basophils: I wonder if pDC also express CD123 in this system and how many basophils were analyzed ? How many events were acquired and accordingly, how was the reduction of basophils calculated ? can you integrate example FACS dot plots into the supplements ? it also looks as if the basophils dropped in a very similar way in the vehicle control group ? can you add an assay to prove specificity against the basophils ?

Reviewer #2

(Remarks to the Author)

Please see the attachment.

Reviewer #3

(Remarks to the Author)

The authors have undertaken a substantial reworking of the manuscript and provided new data. The manuscript is substantially improved as a result. The prior review questions have been addressed and I have no further comments.

Version 2:

Reviewer comments:

Reviewer #1

(Remarks to the Author)

The authors have addressed all comments of the reviewers sufficiently. I suggest to accept the paper in the current draft.

Reviewer #2

(Remarks to the Author)

The manuscript is significantly improved with the additional data and reworded sections.

Please find enclosed a few remaining suggestions to the authors:

Fig. 4A & 4B

Could you precise in the result part at which concentration the compound is active? I suggest to remain consistent in term of unit used ($\mu\text{g}/\text{ml}$ or pM) in the fig 4A vs. 4B (and give the equivalence).

Fig. 6

It would be useful to describe how the AFM28-armed allogeneic NK cells are generated, giving more experimental details on the pre-binding step.

Fig. 7C - Impact on CD16A MFI

It would be complementary to show the impact of the treatment on the % NK cells positive for CD16A. Does the 3G8 antibody used to reveal CD16 expression compete with AFM28 ? The authors should comment on how this potential competition could impact the data

Thanks

Response to the Reviewers

Responses are provided in blue font in sequential order to the specific reviewers' comments.

Reviewer #1:

This is another paper on a bispecific antibody in the setting of AML. The difference to most of the prior reports, is that here an innate bispecific is introduced, that is recruiting NK cells through CD16 engagement. Preclinical data, from *in vitro* to *in vivo* are shown. The data demonstrates efficacy against primary AML and MDS cells and demonstrates efficacy *in vivo* (mouse model) and safety in a cynomolgus monkeys.

The preclinical *in vitro* and *in vivo* data set is sort of "standard" for the regulatory affairs, but for a publication in this journal is missing aspects that are of biological relevance and interest to the scientific community. With the current data set, this is just another paper demonstrating efficacy preclinical, but is not answering any questions on existing or evolving resistance and is not tackling questions of patient derived NK cells or impact of the bone marrow environment on the CD123-CD16a bispecific construct activity. Therefore I like the authors to share some more data on the following aspects:

1. The authors demonstrate the secretion of proinflammatory cytokines by CD123-CD16 activated NK cells - can you please show the impact on the phenotype of NK cells (CD16 downregulation has been reported), but importantly also the impact on AML cells ? e.g. upregulation of HLA class I molecules (inhibitory to NK cells), upregulation of PD-L1 ? The authors need to extend the time of coculture to pick up immune escape mechanisms. How do residual AML cells look after coculture ? is there an outgrowth after removal of the antibody ? At least the impact on the cells within the coculture should be addressed.

We agree with the reviewer that ADCC can have further impact on the NK cell phenotype. We added data on the induction of the NK cell activation marker CD69 in response to AFM28 treatment obtained in our *ex vivo* whole blood assays (revised **Figure 7B**). With regards to CD16 regulation, we added novel data sets, discovering that AFM28 has the potential to inhibit CD16A shedding at saturating concentrations, which was not observed by an Fc-enhanced anti-CD123 IgG1 antibody (revised **Figure 7C and Supplementary Figure 3B-C**). These data were derived from cultures using *ex vivo* whole blood from healthy donors as well as additional *in vitro* experiments using co-cultures of NK cells with CD123+ target cells and PMA/ionomycin for stimulation of CD16A shedding. The following sentences were added to the results.

"AFM28 PD activity in circulating human whole blood cultures was accompanied by increased CD69 expression on NK cells consistent with the activation of NK cells as a result of ADCC. While Fc-mediated targeting of CD16A by IgG1 antibodies is associated with shedding of CD16A, in this study, there was a trend for an increase in CD16A expression on NK cells in the presence of saturating concentrations of AFM28 in whole blood cultures and in response to CD123+ target cells (revised **Figure 7B and Supplementary. Figure 3B**). In this context, it was noted that NK cell exposure to AFM28 could inhibit PMA/ionomycin-induced CD16A shedding but not shedding of CD69 and CD137, which was not observed for an Fc-enhanced anti-CD123 IgG1 antibody (**Supplementary. Figure 3C**). These results suggest that Fc-independent bivalent high-affinity targeting of CD16A by AFM28 can interfere with activation-induced shedding of CD16A and maintain CD16A expression upon ADCC."

The low number of surviving primary leukemic cells in particular after AFM28 exposure in our experimental system precluded investigation of the phenotype of residual primary leukemic cells and/or specific enrichment and longitudinal phenotypic assessment. The authors feel that such

experiments on treatment/ADCC-resistance are certainly of interest, however beyond the scope of this manuscript. However, we concur with the reviewer that IFN-inducible molecules like PD-L1 may be induced as a consequence of ADCC-induced IFN-g (e.g. Gutting et al., Oncoimmunology 2021). In this context, it has been reported that co-culture of non-targeted NK cells with primary leukemic cells resulted in preferential depletion of the leukemic cell fraction positive for the NK cell activating ligands for NKG2D (NKG2D-L), whereas NKG2D-L-negative LSC-like cells survived (Paczulla et al., Nature 2019). Moreover, there was an additional preferential targeting of leukemic cells positive for the ligands CD112 and CD155. At least in the steady-state there were relatively low frequencies of PD-L1-positive AML cells, while it was not investigated whether these are affected after exposure to NK cells (Paczulla et al., Nature 2019).

2. Also, all the experiments were done in an "allo" setting with NK cells from healthy donors. Hence, it would be very interesting to look at i) NK cells from AML patients derived from different time points, e.g. initial diagnosis vs relapse but also ii) in cocultures with complete autologous set up, or just add the construct to primary bone marrow samples.

We would like to thank the reviewer for this suggestion. We have generated two new data sets that suggest that AFM28 can effectively arm AML patients' endogenous NK cells for anti-leukemic activity to a comparable extent to healthy donors at least in some AML patients.

First, we show in an autologous setting using *ex vivo* cultures of prospectively-sampled fresh whole blood, derived from untreated newly diagnosed AML patients (n=6), that AFM28 could demonstrate activity against primary leukemic blasts. Responder samples (n=3) showed a mean lysis of 76% (range 59-90%), while non-responders (n=3) exhibited a mean lysis of 7% (range 0-22%). This novel data set is presented in the **new Figure 4A**. Responsiveness was further enhanced when allogeneic NK cells (used freshly or after *in vitro* expansion and cryopreserved) were added, increasing the proportion of responders (lysis >50%) to 100% (n=6/6) and overall increasing blast lysis to 87% (range 60-100%).

Second, we further explored the ADCC activity in a set of prospectively-sampled patient-derived NK cells, enriched from freshly drawn blood of AML patients (n=5), and tested their ADCC capacity against allogeneic primary leukemic blasts, compared to healthy NK (n=5) cells. Effective blast lysis was achieved by n=3 AML-derived and all n=5 healthy NK cell samples. This novel data set is presented in the **new Figure 4B**.

3. The E:T ratio used is very unphysiological - I wonder if mimicking the E:T ratio in a patient, and demonstrating NK cell expansion over time might better reflect the clinical situation.

We agree that E:T ratios used *in vitro* with allogeneic NK cells harbor the risk of not being translatable to the clinical situation. However, we provide a novel data set in the **new Figure 4**, showing that AFM28 could demonstrate activity against primary leukemic blasts in an autologous setting in *ex vivo* cultures of prospectively-sampled fresh whole blood derived from untreated newly diagnosed AML patients. Responder samples (n=3) showed a mean lysis of 76% (range 59-90%), while non-responders (n=3) exhibited a mean lysis of 7% (range 0-22%). These results indicate the potential to arm AML patients' endogenous NK cells for anti-leukemic activity in a physiological E:T ratio context.

4. The authors need to integrate already running and reported clinical trials with NK-cell recruiting bispecifics as well as CART activities. This is missing in the discussion and also in the introduction,

including Fc optimized anti-CD123 trials, which were applied in conjunction with AZA (to upregulate CD123 and increase NK cell activity), but failed to show any benefit.

We thank the reviewer for this suggestion for improvement. We modified the introduction and discussion, and added several primary references on Fc-enhanced IgG1-based antibodies, IL-3-diphtheria toxin fusion protein and T cell engagers including combinations with hypomethylating agents, novel NK cell engagers, novel CAR-T cell designs, antibody drug conjugates, cytotoxic IL-3 fusion proteins, as single agents or when combined with hypomethylating agents and/or anti-BCL-2 venetoclax.

5. After CD123 CART therapy severe bone marrow aplasia has been observed. Please extend the data set on safety, also considering the fact that IFN γ has been reported to upregulate CD123 on endothelial cells. Hence the monkey model, without any target cells, and hence no secretion of proinflammatory cytokines is not the suitable animal model.

We agree with the reviewer that CD123 targeting entails the risk for bone marrow aplasia. Indeed, assessment of bone marrow aplasia is implemented in the ongoing phase 1 dose-escalation study of AFM28 in R/R AML patients as a potential risk factor.

To expand on preclinical safety parameters, we assessed the colony-forming potential of healthy BMMC-derived CD34+ cells after treatment with AFM28 and allogeneic NK cells, in addition to the already provided data on AML and HR-MDS. In these experiments, we could demonstrate that colony formation of healthy BMMC-CD34+ samples was only modestly affected after exposure to AFM28 (6-39%) compared to the robust reduction in leukemic colony formation of AML (57-75% reduction) and HR-MDS BMMC-CD34+ samples (57-64%). These data are shown in **new Figure 3G**.

Further, we modified the results section to better indicate that cynomolgus monkeys are a relevant animal model for pharmacodynamic and toxicology assessment. AFM28 is species cross-reactivity to cynomolgus CD16, expressed by cynomolgus NK cells, and to cynomolgus CD123, expressed for instance by cynomolgus basophils, resulting in dose-dependent depletion of basophils in peripheral blood of cynomolgus monkeys as described in **Figure 7B**. No adverse clinical observations in cynomolgus were noted.

Altogether, we are confident that the NK cell-dependent mode of action of AFM28 has a better safety profile associated with more transient effects compared to conventional CAR-T cells. Indeed, CAR-T cells are more prone for long-lasting adverse effects due to their proliferative capacity and persistence, with the risk for restricting endogenous recovery from e.g. myelosuppression or on-target/off-tumor effects (Baroni et al., J Immunother Cancer. 2020). This is in line with preclinical data, suggesting that anti-CD123 CAR-NK cells in contrast to anti-CD123 CAR-T cells did not elicit endothelial injury and severe hematopoietic toxicity in a human endothelial murine model and in a humanized mouse model, respectively (Caruso, 2022, Journal of Hematology & Oncology). Moreover, early clinical data of a novel CD123-targeting CAR-T cell design including an on/off switch showed only transient myelosuppression that recovered upon treatment completion (Wermke et al., Blood (2021) 137 (22): 3145–3148). We also included new primary references on anti-CD123 CAR-T cells in the introduction and discussion.

Reviewer #2

Nature Communication paper review A-Summary of Key results: Jana-Julia Siegler et al. describe the preclinical profile of a novel bifunctional NK cell engager (NKCE) molecule targeting CD123 on AML blasts and CD16a on NK cells (AFM28), exhibiting cytotoxic activity against CD123-positive leukemic cells in vitro (with activity in CD64+ cells), against blasts from MDS & AML ex vivo (bone marrow patient samples in presence of healthy donor NK cells) and in vivo in a CD123+ AML mice model. Activity towards AML cell lines irrespective of their levels of CD123, CD64 expression or mutational status & activity towards leukemic stem cells are described. Direct inhibition of IL-3-dependent proliferation of AML cells is shown. PK/PD relationship and a favorable toxicity profile in non-human primates (NHP) are shown, which support clinical development. This molecule differs from other reported CD123-targeting NKCE by the antibody (Ab) format with a tetravalent bispecific CD123/CD16A innate cell engager, bivalent for each target.

B-Validity:

The manuscript is well written, and data clearly presented. C-Originality and significance: The specific CD123-CD16A NKCE represents a novel molecule although the design and therapeutic concept is similar to previously described NKCEs targeting other antigens (doi: 10.1080/19420862.2021.1950264). The generic NKCE platform was previously reported. The therapeutic concept is similar to previously described NKCEs targeting similar antigen (doi: 10.1038/s41587-022-01626-2). The identification of the cytotoxic activity of the NKCE against leukemic stem cells and progenitor cells in AML and MSD is novel.

D- Data & methodology:

Most of the experiments are well designed and data well-presented and robust, based on state-of-the-art methodologies. However, the ex vivo studies are performed using IL2-pretreated healthy donor NK cells as effector cells against patient primary AML cells: evaluating ex vivo activity using an autologous setting would represent experimental conditions more translatable to the clinical situation. A complementary in vivo design using adoptive transfer of human NK cells would complement the analysis of in vivo therapeutic efficacy.

We would like to thank the reviewer for this suggestion. We have generated two new data sets that suggest that AFM28 can effectively arm AML patients' endogenous NK cells for anti-leukemic activity to a comparable extent to healthy donors at least in some AML patients.

First, we show in an autologous setting using *ex vivo* cultures of prospectively-sampled fresh whole blood, derived from untreated newly diagnosed AML patients (n=6), that AFM28 could demonstrate activity against primary leukemic blasts. Responder samples (n=3) showed a mean lysis of 76% (range 59-90%), while non-responders (n=3) exhibited a mean lysis of 7% (range 0-22%). This novel data set is presented in the **new Figure 4A**. Responsiveness was further enhanced when allogeneic NK cells (used freshly or after *in vitro* expansion and cryopreserved) were added, increasing the proportion of responders (lysis >50%) to 100% (n=6/6) and overall increasing blast lysis to 87% (range 60-100%).

Second, we further explored the ADCC activity in a set of prospectively-sampled patient-derived NK cells, enriched from freshly drawn blood of AML patients (n=5), and tested their ADCC capacity against allogeneic primary leukemic blasts, compared to healthy NK (n=5) cells. Effective blast lysis was achieved by n=3 AML-derived and all n=5 healthy NK cell samples. This novel data set is presented in the **new Figure 4B**.

Moreover, we added a new *in vivo* data set on the adoptive transfer of human NK cells armed with AFM28 in irradiated fully immunodeficient hIL-15 NOG mice inoculated IV with EOL-1_Luc cells (**new Figure 6D-E** (previously Figure 5)). Tumor growth was markedly delayed when mice were treated with AFM28-armed NK cells, whereas AFM28 treatment alone failed to induce tumor control in this model due to the lack of responsive effector cells. Adoptive transfer of NK cells alone showed lower anti-tumor activity but only at early time points, suggesting rapid exhaustion of NK cells in the absence of AFM28.

E. Appropriate use of statistics and treatment of uncertainties.

The n of primary AML or MDS samples is low (n=5) and the n of the colony-forming Unit (CFU) assay very low (n=3 AML & n=3 MDS) to draw general conclusions given the heterogeneity of the disease. In Fig 5C, Kaplan-Meier plot shows survival upon treatment, with data represented as mean \pm SEM: the mean \pm SD for the median calculation seems more appropriate.

The number of experiments of the colony-forming unit (CFU) assays were increased to n=5 per indication (AML, MDS and healthy). We have now used at least n=5 primary patient samples per entity for all assays. We would like to emphasize again that these are rare and precious primary patient samples, which makes it difficult to obtain them in general and to obtain enough material to perform as many assays as possible with the same patient material.

The SEM in **Figure 6B** (previously Figure 5) and the **new Figure 6E** was replaced by the SD.

F-Conclusions:

the conclusions and data interpretation are valid and reliable

G-Suggested improvements:

The exact architecture of the NKCE format used should be clarified. It is described as a tetravalent bispecific CD123/CD16A innate cell engager. Does the construct contain an Fc domain? I suggest to add a scheme with relevant details as a supplementary figure.

AFM28 is a tetravalent bispecific humanized IgG1-scFv fusion antibody composed of two CD123 and two CD16A (FcγRIIIA) binding domains (CD123/CD16A innate immune cell engager). AFM28 includes an Fc-silenced portion in the IgG1 backbone that is silenced for binding to FcγR while maintaining binding to the neonatal Fc receptor (FcRn) as was described in the **Supplementary Data**.

We added a scheme of AFM28 and its targeting domains and features to **Supplementary Figure 1**.

Regarding the activity towards leukemic stem cells (LSC), it seems to me important to investigate the potential activity of the compound on normal hematopoietic progenitor cell (HSPC) to see if the compound induces killing of LSC specifically, with normal progenitor capacity maintained (no or minimal HSPC killing?); the documentation of the differential CD123 expression level would be of interest.

We agree that evaluation of the normal healthy bone marrow compartment supports the assessment of the preclinical safety of AFM28. We added new *in vitro* experiments, assessing the colony-forming potential of healthy BMMC-derived CD34+ cells after treatment with AFM28 and allogeneic NK cells, in addition to the already provided data on AML and HR-MDS. In these experiments, we could demonstrate that colony formation of healthy BMMC-CD34+ samples was only modestly affected after exposure to AFM28 (6-39%) compared to the robust reduction in leukemic colony formation of

AML (57-75% reduction) and HR-MDS BMMC-CD34+ samples (57-64%). These data are shown in **new Figure 3G**.

While performing the additional CFU assays, we also determined CD123 expression on the enriched CD34+ cells from healthy donors. On CD34+CD38+ cells, the mean CD123 expression was 4.9% (range: 2.5-8.5%), most comparable to the CD45+CD34+CD38+ leukemic blasts, which were 71.8% (range: 51-84.2%) in AML and 50.1% (range: 30.5-62.4%) in MDS (see **Supplementary Figure 1 C-D**). On CD34+CD38- cells, mean CD123 expression was 4.9% (range: 2.5-8.5%), most comparable to CD45+CD34+CD38-CD117+ LSPCs, which was 71.5% (range: 60.3-91.4%) in AML and 63.4% (range: 33.9-80.1%) in MDS (see **Supplementary Figure 1 C-D**). These data support the assumption that CD123 expression appears to be lower on healthy HSPCs than on leukemic cells, which in turn explains why the number of colonies in the CFU assay in the healthy samples was less reduced by treatment with AFM28 than in the AML and MDS samples.

Evaluating *in vivo* an AML model with low CD123 antigen density would be a plus to confirm *in vitro* findings. The model used is not an AML & its mutational status is not described: an additional human AML or MDS tumor model with a defined mutational status would be better.

Regarding *in vivo* evaluation, the model used is not an AML & its mutational status is not described: an additional human AML tumor model with a defined mutational status or with low CD123 antigen density is missing, to confirm the potential of the molecule to deplete tumor cells irrespective of their mutational profile or CD123 expression level.

We agree with the reviewer that a more physiological approach would be more informative. We feel that the new data set using non-manipulated primary whole blood from different AML patients can provide the most compelling physiologically-relevant context. This novel data set is presented in the **new Figure 4A**. These new data show that AFM28 exposure can elicit anti-leukemic activity in an autologous setting in *ex vivo* cultures of prospectively-sampled fresh whole blood derived from untreated first diagnosis AML patients (n=6) with a variety of mutational profiles (see **Supplementary Table 3**). Responder samples (n=3) showed a mean lysis of 76% (range 59-90%), while non-responders (n=3) exhibited a mean lysis of 7% (range 0-22%). These results suggest the potential to arm the AML patient's endogenous NK cells for anti-leukemic activity.

The *in vivo* models, now further amended in the revised manuscript version in **Figure 6** (previously Figure 5), were used as an adequate model systems to demonstrate the proof-of-concept to address the two questions, (1) if AFM28 can induce tumor growth delay *in vivo* via endogenous innate immune components and (2) if AFM28 can enhance *in vivo* anti-tumor activity of adoptively-transferred human NK cells. The eosinophilic AML cell line EOL-1 was chosen in favor of an in-house established murine C1498 cell line with ectopic human CD123 overexpression, which showed CD123 expression even two to three orders of magnitude higher than detected on AML cells. We would like to state that the EOL-1 cell line faithfully belongs to the rare type of acute myeloid eosinophilic leukemia, which has a KMT2A tandem duplication first described by Saito et al. Blood. Vol 66. No 6 1985: pp 1233-1240 (<https://www.dsmz.de/collection/catalogue/details/culture/ACC-386>). We have added the mutational status of the cell line to the **Supplementary Table 2**. We thank the reviewer for bringing this to our attention.

In Fig. 6D, the increase of IL6 in animals treated with the compound and that develop ADA should be further documented.

We thank the reviewer for making us aware of this issue. We adjusted the order of statements in this section. In this respect, we added that the emerging ADA response in cynomolgus monkey resulted in reduced AFM28 exposure, which coincided with reduced PD activity and modest induction of IL-6 cytokine levels. Further, we also added that the emergence of the ADA response in cynomolgus monkeys, potentially a xenogeneic effect, is not predictive for ADA formation in humans (Swanson and Bussiere, *Current Opinion in Microbiology* 2012).

Also, given that the NKCE concept is already established, the manuscript is missing in mechanistic explorations of the advantage of targeting CD16A selectively as compared to several other CD16-based NKCE that can target CD16A and CD16B.

AFM28 was developed with a proprietary design to specifically redirect the anti-tumor activity of NK cells towards AML cells. Since NK cells only express the activating isoform CD16A and not CD16B, which may have rather an inhibitory role in neutrophils, AFM28 and other innate cell engagers specifically target CD16A. More relevantly in the context of AML, we believe that the benefit of specific CD16A targeting in contrast to pan FcγR targeting is illustrated by the experiments shown in **Figure 1**, demonstrating that a “non-Fc-specific” Fc-enhanced anti-CD123 IgG1 antibody fails to induce ADCC when AML cells express other FcγR. A similar observation has been reported for an NKp46-targeting NK cell engager (Gauthier et al., *Nat. Biotech.* 2023) referenced in this manuscript. Because of the bivalent CD16 targeting, specificity for CD16A is a hallmark of all innate cell engagers to prevent unwanted cross-linking of CD16A+ NK cells with other FcR+ cells including CD16B+ neutrophils and thus to prevent neutrophil killing. Another advantage of the innate cell engager design is the high-affinity bivalent CD16A targeting feature, enabling enhanced NK cell surface retention and reduced competition by serum IgG as compared to previous concepts that were based on lower-affinity monovalent CD16 targeting (Pinto et al., *Trends Immunol.* 2022).

8 AML & 6 MDS samples are described in the supplemental part but the activity data are shown only on 5 samples : could you explained why this is the case?

We thank the reviewer for bringing this uncertainty to our attention. This discrepancy is owing to the fact that we were not able to use the same set of patient samples for all assays due to the limited number and cell count of at least some of the primary patient samples. **Supplementary Tables 3-5** provide a precise breakdown of which sample was used for which assay. To avoid further confusion, we have added the following sentence to the methods section: “Details on the BM and PB donor characteristics and for which assay the samples were used are provided in **Supplementary Tables 3–5.**”

H-References:

Relevant literature is appropriately cited.

I-Clarity and context:

abstract, introduction and conclusions are clearly written and appropriate. The paper is logically structured and well written. The CD123-CD16A NKCE show promising anti-tumor activity with activity on leukemic stem cells and a favorable safety profile. The conclusion that this reagent is ready for clinical testing is supported by the presented data. Inflammatory material: no inappropriate language

Reviewer #3 (Remarks to the Author):

This is a generally well written report on the pre-clinical development of the CD123 innate cell engager AFM28. The experiments are generally well designed and the methods and figures clearly delineated. As acknowledged by the authors CD123 is an attractive target for leukaemia-directed therapy with a view to directly eliminating LSC populations and/or interrupting IL3 trophic signalling. Despite its attractiveness, effective clinical targeting of CD123 with various agents has proven elusive. In the current paper Siegler and colleagues has described the next logical step in the development of CD123-directed therapy with the development of an NK-engaging tetra-valent molecule. The experiments that are outlined in the paper are a logical sequence of pre-clinical work up of a new drug but overall do not reveal any particularly new findings into the therapeutic application of this ICE.

The sensitivity of AML cells to allogeneic NK cells in the presence of AFM28 is a reasonable experimental design, but would have been more clinically relevant if performed using NK cells autologous to the AML targets. Presumably the samples containing leukaemic targets would also contain some residual NK populations. The confounding effect of HLA-disparity/non-self recognition of leukaemic targets by allogeneic NK cells limits the interpretation of this finding.

We would like to thank the reviewer for this suggestion. We provided a novel data set showing that AFM28 could demonstrate activity against primary leukemic blasts in an autologous setting in *ex vivo* cultures of prospectively-sampled fresh whole blood derived from untreated newly diagnosed AML patients (n=6). Responder samples (n=3) showed a mean lysis of 76% (range 59-90%), while non-responders (n=3) exhibited a mean lysis of 7% (range 0-22%). This novel data set is presented in the **new Figure 4A**. Responsiveness was further enhanced when allogeneic NK cells (used freshly or after *in vitro* expansion and cryopreserved) were added, increasing the proportion of responders (lysis >50%) to 100% (n=6/6) and overall increasing blast lysis to 87% (range 60-100%).

Importantly, the authors discuss the role of NK cell fitness in AML patients and its potentially limiting effect in ICE-based therapies. Whilst there is generally (sufficiently) preserved NK cell numbers and function in patients with newly diagnosed AML, further consideration and discussion should be undertaken regarding how this may change following chemotherapy delivery and/or in the setting of relapsed disease, which is likely to reflect the population in which these novel agents are to be initially trialed.

In addition to **new Figure 4A**, we further explored the ADCC activity of patient-derived NK cells, enriched from freshly drawn blood of AML patients (n=5) and tested against allogeneic primary leukemic blasts of the same AML patient, compared to healthy NK (n=5) cells. This novel data set is presented in the **new Figure 4B**. Effective blast lysis was achieved by n=3 AML-derived and all n=5 healthy NK cell samples. Collectively, these results suggest that AFM28 can effectively arm AML patients' endogenous NK cells for anti-leukemic activity to a comparable extent to healthy donors at least in some AML patients.

Moreover, we added available primary references to the discussion, describing the status of NK cells of AML patients after relapse, after chemotherapy and after complete remission in the context of the potential for AFM28 efficacy as a single agent or in combination with allogeneic NK cell therapy.

Further consideration and discussion should be undertaken as to the expression of inhibitory NK receptor ligands on LSCs targets, which may limit NK cell activation *in vivo*.

The authors would like to thank the reviewer for this recommendation. Generally, there is only few knowledge on the expression of NK cell inhibitory ligands as well as NK cell activating ligands. Most prominently it has been shown that LSC-like cells express reduced levels of NKG2D ligands, which has been suggested to support immune evasion from NK cells (Paczulla et al., Nature 2019). Further these authors reported high frequencies of AML cells positive for the NK cell activating ligands CD112 and CD155, whereas low frequencies were positive for B7-H6 and PD-L1. Moreover, immunomodulating molecules such as the NK cell receptors CD44, CD96 and TIM-3 have been described to be expressed on AML cells including LSC-like cells (Inagaki Y et al., Cell Stem Cell, 2010; Stelmach and Trumpp, Haematologica 2023).

We amended the discussion as follows: “It remains to be investigated whether additional targeting of activating and inhibitory NK cell ligands expressed by leukemic blasts including LSC-like cells (Stelmach and Trumpp, Haematologica 2023) enhances ADCC to further restrict the risk for minimal residual disease.”

Response to the reviewers comment to the revised manuscript

Responses are provided in blue font in sequential order to the specific reviewers' comments.

Reviewer #1:

Thank you for the revision and integrating data of additional experiments. I have 3 questions / comments to the presented novel data sets:

1. While CD69 is frequently used as an early activation marker for NK cells, it has significant limitations in terms of specificity and functional correlation. CD69 is an early and transient marker that does not necessarily correlate with functional cytotoxicity or sustained NK cell activation. This makes it a less reliable indicator of specific NK cell-target interactions unlike markers such as CD107a (degranulation) or IFN- γ production.

Thank you for this advice. We added data on NK cell degranulation and IFN-gamma induction in response to target cells and AFM28 to the revised **Supplementary Figure 1C**.

To amend our data, showing that induction of NK cell activation markers associated with function, we added a data set, depicting the upregulation of the later-stage NK cell activation markers CD137 and CD25 in response to target cells and AFM28, which coincided with the depletion of the target cells. Data are added to the revised **Supplementary Figure 1D**. Likewise, up-regulation of CD137 and CD25 on NK cells coincided with basophil depletion. Data are added to the revised **Figure 7B** and revised **Supplementary Figure 5A**. The corresponding result sections were amended accordingly.

We are convinced that these data collectively corroborate that up-regulation of NK cell activation markers, coinciding with NK cell-mediated depletion of CD123⁺ target cells, are adequate surrogate markers of NK cell function.

2. Looking at the cytokine profile it is hard to believe that IFN γ and TNF do not compromise activity, as there is a high amount secreted and it is known and published, that these cytokines lead to upregulation of inhibitory checkpoint molecules. Most likely the assay time is too short to assess this adaptive resistance factor, however in the patient this is likely to occur. Can you look at the AML blasts, e.g. in a less favorable E:T ratio and check for upregulation after e.g. 72 hours ?

Thank you for the clarification and suggestion, which certainly will provide new research opportunities for future preclinical research and clinical combination studies, despite deviating from the main scope of this manuscript.

As suggested, we additionally investigated the up-regulation of inhibitory molecules like PD-L1 and HLA-I on AML blasts after an extended 48 h co-culture with allogeneic NK cells. While strong anti-leukemic activity was maintained in this setting, exposure to allogeneic NK cells led to an increase in PD-L1 expression on the remaining leukemic blasts, albeit to very heterogeneous extent between the AML samples, with a trend for a further increase by AFM28. Minor effects on the regulation of HLA-I were observed. Data are added to the revised **Supplementary Figure 4**, and the corresponding result section was amended accordingly. We added to the discussion that it remains to be explored in future preclinical studies whether baseline or therapy-induced PD-L1 affects the activity of cellular immunotherapies in AML, while on the other hand providing opportunities for potential combinations of AFM28 with PD-1 negative NK cell products or PD-1/PD-L1 checkpoint therapy.

3. Basophils: I wonder if pDC also express CD123 in this system and how many basophils were analyzed ? How many events were acquired and accordingly, how was the reduction of basophils calculated ? can you integrate example FACS dot plots into the supplements ? it also looks as if the

basophils dropped in a very similar way in the vehicle control group ? can you add an assay to prove specificity against the basophils ?

We agree that pDC are known for CD123 expression and present a rationale target analogous to basophils. To better illustrate and elaborate this point, we added a data set, showing both the depletion of basophils and pDC from leukocyte cultures after AFM28 exposure, depicting the percentage change from baseline, raw counts and example FACS dot plots (including CD123 expression for pDC), alongside the upregulation of the NK cell activation markers CD137 and CD25. These data are added to the revised **Supplementary Figure 5B** and **Figure 7B** as well as to the corresponding result section.

Regarding your second question, please note that the reduction in basophils was more pronounced in AFM28-treated animals and was sustained for two to four weeks, whereas the vehicle-treated control group showed only very transient, more moderate effects that recovered within one week. We clarified the corresponding result section. We added that AFM28-treated animals also showed pDC reduction that was sustained for at least two weeks, whereas vehicle-treated animals showed only modest and very transient effects. These data are shown in the revised **Supplementary Figure 5C** as well as in the corresponding result section. We clarified the calculation of basophil reduction in the corresponding method section as follows: "Absolute cell counts, that had been computed from relative percentages determined by flow cytometry and absolute leukocyte counts determined by manual leukocyte differential of blood smears, were normalized to day 0 and depicted as percentage of baseline."

Regarding your last question about CD123 target specificity, we clarified the result section of **Figure 1A+B** that showed no lysis of CD123-negative cells in the presence of AFM28. To corroborate its specificity against CD123⁺ target cells, we added new data demonstrating that bystander CD123-negative (CD30-positive) tumor cells are not killed in the presence of CD123-positive target cells and AFM28. However, these cells were susceptible to ADCC mediated by a CD30/CD16A bispecific engager, further corroborating the previous data. These data are depicted in the revised **Supplementary Figure 1B**. We believe that these data show specificity of AFM28-induced ADCC towards CD123⁺ cells, in line with the clarification of sustained basophil reduction in AFM28-treated cynomolgus animals as compared to the transient basophil reduction in vehicle-treated animals.

Reviewer #2:

Thanks for adding the gating strategy of blasts and LSPCs, and the proportion of CD123⁺ cells of AML and HR-MDS patient-derived BMMC samples. I would not put these data in the same supp fig as the AFM8 structure but to a distinct one since this is not the same topic.

I don't see the CD123 expression on the CD34⁺ cells from healthy donors on Supp Fig1? It is I guess on Fig3G in main part of the manuscript? It would be useful to get the pictures of the CFU plates and a table with the CFU manual counts. Thanks.

Thank you for making us aware of the missing data piece. We added the CD123 expression of the CD34⁺ CD38⁺ and CD34⁺ CD38⁻ cells from healthy donors to the revised **Supplementary Figure 2D**. As advised, we have moved the gating strategy to the revised **Supplementary Figure 2A**. Moreover, we added CFU manual counts to the revised **Supplementary Table 7**.

Thanks for adding these data addressing the different points raised. Regarding Fig 4, it is important to precise the E:T ratio in the autologous setting without addition of NK cells, or with addition of allogenic NK cells, in order to compare the cytotoxic activities measured at 24h, since the E:T ratio will impact the level of activity. Also the allogenic NK cells added are non expanded fresh ones or expanded frozen ones. Did you compare the phenotype of these different NK cells in term of basal

activation? The authors should explain in the material and methods the protocol for expansion of NK cells and the phenotypes of the NK cells (hyperactivated?) post-expansion.

Thank you for making us aware to clarify these points in the manuscript. We implemented your comments as follows:

As requested, we added the E:T ratios of the autologous setting within the figure legend of **Figure 4** and amended the corresponding result section. Moreover, we included in the figure legend that allogeneic NK cells were added at a 1:1 allogeneic NK cells to peripheral leukocytes ratio. Furthermore, we specified in the figure legend the usage of non-expanded, fresh allogeneic NK cells and expanded, cryopreserved allogeneic NK cells. We clarified and updated the corresponding supplementary method section which describes the protocol for NK cell expansion. Here, we added information on the phenotype of the expanded NK cells, stating that cytokine-expanded NK cells displayed an activated phenotype as indicated by increased expression of CD56, NKp44 and NKp30, while initial induction of the early activation marker CD69 diminished over time.

Moreover, we amended and clarified in the corresponding result section that, as depicted in **Figure 4**, on average, basal anti-leukemic activity of the NK cells was relatively comparable between non-expanded, fresh NK cells and the cytokine-expanded, cryopreserved NK cells against the tested AML samples, likely attributed to the freezing/thawing procedure of the cytokine-expanded NK cells (Damodharan et al., *Cytotherapy* 2020, DOI: 10.1016/j.jcyt.2020.05.001; Mark et al., *Nat. Commun.* 2020, DOI: 10.1038/s41467-020-19094-0).

AFM28 treatment at 3 mg/kg alone failed to induce tumor control in the NSG mice model with adoptive transfer of human NK (activity only in mice treated with AFM28-armed NK cells). Did the authors try different amounts of human NK cells? Did they evaluate different NK cell batches given the variability between healthy donors? The material & method part is not detailed enough for this model.

Thanks for the comment to amend the method sections. We added more detailed information on the NK cell batch preparation to the corresponding method sections.

As previously advised, we had added a complimentary *in vivo* CDX model with adoptive transfer of human NK cells with AFM28 to the manuscript. These results demonstrate the proof-of-concept of *in vivo* efficacy for the therapeutic combination of cytokine-expanded allogeneic NK cells and AFM28. Since a large number of NK cells was needed for this proof-of-concept *in vivo* experiment (i.e., at least 6×10^8 NK cells), we had used one available cryopreserved batch/production of cytokine-expanded allogeneic NK cells (derived from one healthy donor) at one concentration.

As we adhere to the 3R's of animal experimentation, we believe that for the scope of this manuscript, it is ethically justifiable to perform this proof-of-concept *in vivo* experiment for AFM28 with one batch of NK cells. We show throughout the manuscript that NK cells from different donors universally respond to AFM28 *in vitro*. In addition, this NK cell batch had also been included in **Figure 4A**, showing anti-leukemic activity that was relatively comparable to that of the four fresh, non-expanded buffy coat-derived NK cell preparations.

As part of this rebuttal, to support our statement, while stating that these data should not be published, we share an overview of six previously produced NK cell expansions (i.e., derived from six different healthy donors) using the same IL-2/IL-15-based expansion protocol as described in this manuscript. All preparations were subjected to standardized testing of phenotype and ADCC activity (using the CD30/CD16A engager AFM13 against CD30+ target cells) after 1 day vs 14 days of expansion. All NK cell expansions, including the one used in this manuscript (red data points), showed an activated phenotype after 14 days exposure as indicated by increased expression of CD56, NKp44 and NKp30, while initial induction of the early activation marker CD69 diminished over time (see figure below). All NK cell preparations showed ADCC activity.

Notably, one of these additional preparations (blue data points) was tested in another proof-of-concept *in vivo* CDX model, demonstrating increased anti-tumor activity against CD30⁺ target cells in combination with the CD30/CD16A engager AFM13 (see figure below) analogous to the CD123⁺ CDX model for AFM28 in this manuscript.

We agree that if the scope was the clinical development of an off-the-shelf NK cell product, it would be necessary to assess batch-to-batch variability across preparations, manufacturing runs and donor history. Clinical application of such a NK cell product would include testing of the therapeutically effective dose. For the scope of this manuscript, we believe that the use of one batch of NK cells is sufficient to demonstrate the proof-of-concept of AFM28 with NK cells.

[FIGURE REDACTED]

The authors state that “Adoptive transfer of NK cells alone showed limited anti-leukemic activity and only at early time points in this model, suggesting rapid exhaustion of NK cells in the absence of AFM28”: It is not clear if these observations were reproduced on at least 2 to 3 NK batches. If they is only one NK donor evaluated, no conclusion can be made.

We agree that this statement is speculative and removed it from the manuscript.

Thanks for separating the ADA+ or - NHPs.

Fig7B shows the activity of the compound in whole blood (WB) in term of basophil depletion and the CD16a expression on NK cells in WB: to interpret the data, it would be useful to know the NK:blast ratio in these samples and to correlate the observations of CD16a expression level modulations with the activity of the compound (basophil depletion). In supplementary Figure 3B showing CD16a decrease at 0.1 µg/ml but an increase at 1 µg/ml: I would recommend to organize the data

separating 3h & 24h since the E:T ratio of 1:1 does not give the same level of activity at 3h with this low E:T ratio and at 24h. It is important to read the modulation of CD16a expression at an active dose vs a poorly active one: the data of cytotoxic activity at these 2 time points are missing.

Thank you for the suggestion to improve the manuscript for these points. We added the range of E:T cell ratios of NK cells to basophils in the WB assay to the corresponding legend of the revised **Figure 7**. There was no correlation between the extent of basophil depletion and the E:T ratio.

Moreover, we added the missing cytotoxicity data to the revised **Supplementary Figure 6A**. These data corroborate the PMA/ionomycin data of the revised **Supplementary Figure 5B**, suggesting that CD16A modulation is concentration-dependent and CD16A-engagement-dependent and not necessarily correlated with cytotoxic activity. We plan to dedicate further clarification of the mechanisms of action for AFM28-mediated inhibition of CD16A shedding to a more focused, separate future study.

Thanks for these explanations. Comparisons of CD16a specific targeting versus a FcγR targeting with a Fc moiety *in vivo* would be useful to confirm these statements.

Thank you for suggesting to show the comparison between CD16A-specific targeting vs FcγR-targeting in a more stringent experimental setting with physiological relevance. We added a data set of primary bone marrow samples from n=10 AML patients, demonstrating that the depletion of leukemic blasts by AFM28 was more potent as compared to that by an Fc-enhanced anti-CD123 IgG1 antibody. Moreover, this differentiation appeared to be increased in the case of targeting leukemic blasts with high CD64 expression as compared to targeting leukemic blasts with low CD64 expression, corroborating the results of the cell lines presented in **Figure 1**. Data are added to **Figure 2E-H** and **Supplementary Figure 2E** and described in the corresponding result section. We believe that these data are more valuable since this model accounts for the heterogeneity of leukemic blasts as compared to *in vivo* data from cell lines.

We would like to mention that the clinical development of the Fc-enhanced anti-CD123 IgG1 antibody talacotuzumab has been terminated due to an unfavorable risk/benefit profile in spite of anti-tumor activity in preclinical models (Kubasch et al., *Leukemia* 2020, DOI: 10.1038/s41375-019-0645-z; Montesinos et al., *Leukemia* 2021, DOI: 10.1038/s41375-020-0773-5).

In contrast, as reported at last year's conference of the American Society of Hematology in an oral presentation, AFM28 monotherapy has shown early signs of clinical activity and a well-managed safety profile in an ongoing dose-escalation phase 1 study in heavily pretreated R/R AML patients (Montesinos et al., *Blood* 2024, DOI: 10.1182/blood-2024-194356). Hence, based on the available preclinical and clinical data, we feel that an additional *in vivo* CDX study is not ethically justifiable.

Response to the Reviewers

Responses are provided in blue font in sequential order to the specific reviewers' comments.

Reviewer #1 (Remarks to the Author):

The authors have addressed all comments of the reviewers sufficiently. I suggest to accept the paper in the current draft.

Reviewer #2 (Remarks to the Author):

The manuscript is significantly improved with the additional data and reworded sections.

Please find enclosed a few remaining suggestions to the authors:

Fig. 4A & 4B

Could you precise in the result part at which concentration the compound is active? I suggest to remain consistent in term of unit used ($\mu\text{g}/\text{ml}$ or pM) in the fig 4A vs. 4B (and give the equivalence).

Overall, AFM28 achieved maximal ADCC activity at concentrations of approx. 100 pM. We added this information to the summary in the discussion. In Fig. 4A/B, we converted the displayed concentrations to pM. 1 $\mu\text{g}/\text{mL}$ are equivalent to approx. 5000 pM, 0.01 $\mu\text{g}/\text{mL}$ are equivalent to approx. 50 pM. We added the molecular weight of AFM28 (203 kDa) to the Supplementary Methods. For overall consistency, we converted the concentrations throughout the figures and the manuscript into pM.

Fig. 6

It would be useful to describe how the AFM28-armed allogeneic NK cells are generated, giving more experimental details on the pre-binding step.

As suggested, we added a detailed description of the pre-binding step to the corresponding method section ("NK cell expansion") in the Supplement.

Fig. 7C - Impact on CD16A MFI

It would be complementary to show the impact of the treatment on the % NK cells positive for CD16A. Does the 3G8 antibody used to reveal CD16 expression compete with AFM28? The authors should comment on how this potential competition could impact the data

We added to the corresponding figure legend that the 3G8 antibody used to reveal CD16 expression does not compete for CD16A binding with innate cell engagers like AFM28. This observation is particularly illustrated by the data presented in Supplementary Figure 6, showing increased (panel A) and maintained (panel B) 3G8-detected CD16 expression intensities on NK cells decorated with saturating concentrations of AFM28. Instead, competition would have been indicated by decreased 3G8-detected CD16 expression intensities. We added that these data are consistent with our previous publication (Pahl et al., Cancer Immunol Res. 2018, PMID: 29514797), showing that 3G8 did not compete with another innate cell engager acimtamig (AFM13) targeting the same CD16A epitope.

Thanks

Version 0: Reviewer #2 attachment

Nature Communication paper review

A-Summary of Key results:

Jana-Julia Siegler et al. describe the preclinical profile of a novel bifunctional NK cell engager (NKCE) molecule targeting CD123 on AML blasts and CD16a on NK cells (AFM28), exhibiting cytotoxic activity against CD123-positive leukemic cells in vitro (with activity in CD64+ cells), against blasts from MDS & AML ex vivo (bone marrow patient samples in presence of healthy donor NK cells) and in vivo in a CD123+ AML mice model. Activity towards AML cell lines irrespective of their levels of CD123, CD64 expression or mutational status & activity towards leukemic stem cells are described. Direct inhibition of IL-3-dependent proliferation of AML cells is shown. PK/PD relationship and a favorable toxicity profile in non-human primates (NHP) are shown, which support clinical development. This molecule differs from other reported CD123-targeting NKCE by the antibody (Ab) format with a tetravalent bispecific CD123/CD16A innate cell engager, bivalent for each target.

B-Validity: The manuscript is well written, and data clearly presented.

C-Originality and significance:

The specific CD123-CD16A NKCE represents a novel molecule although the design and therapeutic concept is similar to previously described NKCEs targeting other antigens (doi: 10.1080/19420862.2021.1950264).

The generic NKCE platform was previously reported. The therapeutic concept is similar to previously described NKCEs targeting similar antigen (doi: 10.1038/s41587-022-01626-2).

The identification of the cytotoxic activity of the NKCE against leukemic stem cells and progenitor cells in AML and MSD is novel.

D- Data & methodology:

Most of the experiments are well designed and data well-presented and robust, based on state-of-the-art methodologies.

However, the ex vivo studies are performed using IL2-pretreated healthy donor NK cells as effector cells against patient primary AML cells: evaluating ex vivo activity using an autologous setting would represent experimental conditions more translatable to the clinical situation.

A complementary in vivo design using adoptive transfer of human NK cells would complement the analysis of in vivo therapeutic efficacy.

E. Appropriate use of statistics and treatment of uncertainties.

The n of primary AML or MDS samples is low (n=5) and the n of the colony-forming Unit (CFU) assay very low (n=3 AML & n=3 MDS) to draw general conclusions given the heterogeneity of the disease.

In Fig 5C, Kaplan-Meier plot shows survival upon treatment, with data represented as mean \pm SEM: the mean \pm SD or the median calculation seems more appropriate.

F-Conclusions: the conclusions and data interpretation are valid and reliable

G-Suggested improvements:

The exact architecture of the NKCE format used should be clarified. It is described as a tetravalent bispecific CD123/CD16A innate cell engager. Does the construct contain an Fc domain? I suggest to add a scheme with relevant details as a supplementary figure.

Regarding the activity towards leukemic stem cells (LSC), it seems to me important to investigate the potential activity of the compound on normal hematopoietic progenitor cell (HSPC) to see if the compound induces killing of LSC specifically, with normal progenitor capacity maintained (no or minimal HSPC killing?); the documentation of the differential CD123 expression level would be of interest.

Evaluating in vivo an AML model with low CD123 antigen density would be a plus to confirm in vitro findings. The model used is not an AML & its mutational status is not described: an additional human AML or MDS tumor model with a defined mutational status would be better.

In Fig. 6D, the increase of IL6 in animals treated with the compound and that develop ADA should be further documented.

Also, given that the NKCE concept is already established, the manuscript is missing in mechanistic explorations of the advantage of targeting CD16A selectively as compared to several other CD16-based NKCE that can target CD16A and CD16B.

8 AML & 6 MDS samples are described in the supplemental part but the activity data are shown only on 5 samples : could you explained why this is the case?

H-References: Relevant literature is appropriately cited.

I-Clarity and context: abstract, introduction and conclusions are clearly written and appropriate. The paper is logically structured and well written. The CD123-CD16A NKCE show promising anti-tumor activity with activity on leukemic stem cells and a favorable safety profile. The conclusion that this reagent is ready for clinical testing is supported by the presented data. Inflammatory material: no inappropriate language

Nature Communication paper review

A-Summary of Key results:

Jana-Julia Siegler et al. describe the preclinical profile of a novel bifunctional NK cell engager (NKCE) molecule targeting CD123 on AML blasts and CD16a on NK cells (AFM28), exhibiting cytotoxic activity against CD123-positive leukemic cells in vitro (with activity in CD64+ cells), against blasts from MDS & AML ex vivo (bone marrow patient samples in presence of healthy donor NK cells) and in vivo in a CD123+ AML mice model. Activity towards AML cell lines irrespective of their levels of CD123, CD64 expression or mutational status & activity towards leukemic stem cells are described. Direct inhibition of IL-3-dependent proliferation of AML cells is shown. PK/PD relationship and a favorable toxicity profile in non-human primates (NHP) are shown, which support clinical development. This molecule differs from other reported CD123-targeting NKCE by the antibody (Ab) format with a tetravalent bispecific CD123/CD16A innate cell engager, bivalent for each target.

B-Validity:

The manuscript is well written, and data clearly presented.

C-Originality and significance:

The specific CD123-CD16A NKCE represents a novel molecule although the design and therapeutic concept is similar to previously described NKCEs targeting other antigens (doi: 10.1080/19420862.2021.1950264). The generic NKCE platform was previously reported.

The therapeutic concept is similar to previously described NKCEs targeting similar antigen (doi: 10.1038/s41587-022-01626-2).

The identification of the cytotoxic activity of the NKCE against leukemic stem cells and progenitor cells in AML and MSD is novel.

Data & methodology: Most of the experiments are well designed and data well-presented and robust, based on state-of-the-art methodologies.

However, the ex vivo studies are performed using IL2-pretreated healthy donor NK cells as effector cells against patient primary AML cells: evaluating ex vivo activity using an autologous setting would represent experimental conditions more translatable to the clinical situation.

A complementary in vivo design using adoptive transfer of human NK cells would complement the analysis of in vivo therapeutic efficacy.

D. Appropriate use of statistics and treatment of uncertainties.

The n of primary AML or MDS samples is low (n=5) and the n of the colony-forming Unit (CFU) assay

very low (n=3 AML & n=3 MDS) to draw general conclusions given the heterogeneity of the disease.

2

Internal

In Fig 5C, Kaplan-Meier plot shows survival upon treatment, with data represented as mean \pm SEM: the

mean \pm SD or the median calculation seems more appropriate.

E-Conclusions: the conclusions and data interpretation are valid and reliable

F-Suggested improvements:

Regarding in vivo evaluation, the model used is not an AML & its mutational status is not described: an

additional human AML tumor model with a defined mutational status or with low CD123 antigen density is missing, to confirm the potential of the molecule to deplete tumor cells irrespective of their mutational profile or CD123 expression level.

In Fig. 6D, the increase of IL6 in animals treated with the compound and that develop ADA should be further documented.

Also, given that the concept is already established, the manuscript is missing in mechanistic explorations of the advantage of targeting CD16A selectively as compared to other CD16-based NKCE that can target CD16A and CD16B.

8 AML & 6 MDS samples are described in the supplemental part but the activity data are shown only on 5 samples : could you explained why this is the case?

The exact architecture of the NKCE format used should be clarified. It is described as a tetravalent bispecific CD123/CD16A innate cell engager. Does the construct contain an Fc domain? I suggest to add a cartoon with relevant details as a supplementary Figure.

G-References: Relevant literature is appropriately cited.

H-Clarity and context: abstract, introduction and conclusions are clearly written and appropriate. The paper is logically structured and well written. The CD123-CD16A NKCE show promising anti-tumor activity with activity on leukemic stem cells and a favorable safety profile. The conclusion that this reagent is ready for clinical testing is supported by the presented data.

Inflammatory material: no inappropriate language

Response to the Reviewers

Responses are provided in blue font in sequential order to the specific reviewers' comments.

Reviewer #1:

This is another paper on a bispecific antibody in the setting of AML. The difference to most of the prior reports, is that here an innate bispecific is introduced, that is recruiting NK cells through CD16 engagement. Preclinical data, from in vitro to in vivo are shown. The data demonstrates efficacy against primary AML and MDS cells and demonstrates efficacy in vivo (mouse model) and safety in a cynomolgus monkeys.

The preclinical in vitro and in vivo data set is sort of "standard" for the regulatory affairs, but for a publication in this journal is missing aspects that are of biological relevance and interest to the scientific community. With the current data set, this is just another paper demonstrating efficacy preclinical, but is not answering any questions on existing or evolving resistance and is not tackling questions of patient derived NK cells or impact of the bone marrow environment on the CD123-CD16a bispecific construct activity. Therefore I like the authors to share some more data on the following aspects:

1. The authors demonstrate the secretion of proinflammatory cytokines by CD123-CD16 activated NK cells - can you please show the impact on the phenotype of NK cells (CD16 downregulation has been reported), but importantly also the impact on AML cells ? e.g. upregulation of HLA class I molecules (inhibitory to NK cells), upregulation of PD-L1 ? The authors need to extend the time of coculture to pick up immune escape mechanisms. How do residual AML cells look after coculture ? is there an outgrowth after removal of the antibody ? At least the impact on the cells within the coculture should be addressed.

We agree with the reviewer that ADCC can have further impact on the NK cell phenotype. We added data on the induction of the NK cell activation marker CD69 in response to AFM28 treatment obtained in our *ex vivo* whole blood assays (revised **Figure 7B**). With regards to CD16 regulation, we added novel data sets, discovering that AFM28 has the potential to inhibit CD16A shedding at saturating concentrations, which was not observed by an Fc-enhanced anti-CD123 IgG1 antibody (revised **Figure 7C and Supplementary Figure 3B-C**). These data were derived from cultures using *ex vivo* whole blood from healthy donors as well as additional *in vitro* experiments using co-cultures of NK cells with CD123+ target cells and PMA/ionomycin for stimulation of CD16A shedding. The following sentences were added to the results.

"AFM28 PD activity in circulating human whole blood cultures was accompanied by increased CD69 expression on NK cells consistent with the activation of NK cells as a result of ADCC. While Fc-mediated targeting of CD16A by IgG1 antibodies is associated with shedding of CD16A, in this study, there was a trend for an increase in CD16A expression on NK cells in the presence of saturating concentrations of AFM28 in whole blood cultures and in response to CD123+ target cells (revised **Figure 7B and Supplementary. Figure 3B**). In this context, it was noted that NK cell exposure to AFM28 could inhibit PMA/ionomycin-induced CD16A shedding but not shedding of CD69 and CD137, which was not observed for an Fc-enhanced anti-CD123 IgG1 antibody (**Supplementary. Figure 3C**). These results suggest that Fc-independent bivalent high-affinity targeting of CD16A by AFM28 can interfere with activation-induced shedding of CD16A and maintain CD16A expression upon ADCC."

The low number of surviving primary leukemic cells in particular after AFM28 exposure in our experimental system precluded investigation of the phenotype of residual primary leukemic cells and/or specific enrichment and longitudinal phenotypic assessment. The authors feel that such

experiments on treatment/ADCC-resistance are certainly of interest, however beyond the scope of this manuscript. However, we concur with the reviewer that IFN-inducible molecules like PD-L1 may be induced as a consequence of ADCC-induced IFN-g (e.g. Gutting et al., Oncoimmunology 2021). In this context, it has been reported that co-culture of non-targeted NK cells with primary leukemic cells resulted in preferential depletion of the leukemic cell fraction positive for the NK cell activating ligands for NKG2D (NKG2D-L), whereas NKG2D-L-negative LSC-like cells survived (Paczulla et al., Nature 2019). Moreover, there was an additional preferential targeting of leukemic cells positive for the ligands CD112 and CD155. At least in the steady-state there were relatively low frequencies of PD-L1-positive AML cells, while it was not investigated whether these are affected after exposure to NK cells (Paczulla et al., Nature 2019).

2. Also, all the experiments were done in an "allo" setting with NK cells from healthy donors. Hence, it would be very interesting to look at i) NK cells from AML patients derived from different time points, e.g. initial diagnosis vs relapse but also ii) in cocultures with complete autologous set up, or just add the construct to primary bone marrow samples.

We would like to thank the reviewer for this suggestion. We have generated two new data sets that suggest that AFM28 can effectively arm AML patients' endogenous NK cells for anti-leukemic activity to a comparable extent to healthy donors at least in some AML patients.

First, we show in an autologous setting using *ex vivo* cultures of prospectively-sampled fresh whole blood, derived from untreated newly diagnosed AML patients (n=6), that AFM28 could demonstrate activity against primary leukemic blasts. Responder samples (n=3) showed a mean lysis of 76% (range 59-90%), while non-responders (n=3) exhibited a mean lysis of 7% (range 0-22%). This novel data set is presented in the **new Figure 4A**. Responsiveness was further enhanced when allogeneic NK cells (used freshly or after *in vitro* expansion and cryopreserved) were added, increasing the proportion of responders (lysis >50%) to 100% (n=6/6) and overall increasing blast lysis to 87% (range 60-100%).

Second, we further explored the ADCC activity in a set of prospectively-sampled patient-derived NK cells, enriched from freshly drawn blood of AML patients (n=5), and tested their ADCC capacity against allogeneic primary leukemic blasts, compared to healthy NK (n=5) cells. Effective blast lysis was achieved by n=3 AML-derived and all n=5 healthy NK cell samples. This novel data set is presented in the **new Figure 4B**.

3. The E:T ratio used is very unphysiological - I wonder if mimicking the E:T ratio in a patient, and demonstrating NK cell expansion over time might better reflect the clinical situation.

We agree that E:T ratios used *in vitro* with allogeneic NK cells harbor the risk of not being translatable to the clinical situation. However, we provide a novel data set in the **new Figure 4**, showing that AFM28 could demonstrate activity against primary leukemic blasts in an autologous setting in *ex vivo* cultures of prospectively-sampled fresh whole blood derived from untreated newly diagnosed AML patients. Responder samples (n=3) showed a mean lysis of 76% (range 59-90%), while non-responders (n=3) exhibited a mean lysis of 7% (range 0-22%). These results indicate the potential to arm AML patients' endogenous NK cells for anti-leukemic activity in a physiological E:T ratio context.

4. The authors need to integrate already running and reported clinical trials with NK-cell recruiting bispecifics as well as CART activities. This is missing in the discussion and also in the introduction,

including Fc optimized anti-CD123 trials, which were applied in conjunction with AZA (to upregulate CD123 and increase NK cell activity), but failed to show any benefit.

We thank the reviewer for this suggestion for improvement. We modified the introduction and discussion, and added several primary references on Fc-enhanced IgG1-based antibodies, IL-3-diphtheria toxin fusion protein and T cell engagers including combinations with hypomethylating agents, novel NK cell engagers, novel CAR-T cell designs, antibody drug conjugates, cytotoxic IL-3 fusion proteins, as single agents or when combined with hypomethylating agents and/or anti-BCL-2 venetoclax.

5. After CD123 CART therapy severe bone marrow aplasia has been observed. Please extend the data set on safety, also considering the fact that IFN γ has been reported to upregulate CD123 on endothelial cells. Hence the monkey model, without any target cells, and hence no secretion of proinflammatory cytokines is not the suitable animal model.

We agree with the reviewer that CD123 targeting entails the risk for bone marrow aplasia. Indeed, assessment of bone marrow aplasia is implemented in the ongoing phase 1 dose-escalation study of AFM28 in R/R AML patients as a potential risk factor.

To expand on preclinical safety parameters, we assessed the colony-forming potential of healthy BMMC-derived CD34+ cells after treatment with AFM28 and allogeneic NK cells, in addition to the already provided data on AML and HR-MDS. In these experiments, we could demonstrate that colony formation of healthy BMMC-CD34+ samples was only modestly affected after exposure to AFM28 (6-39%) compared to the robust reduction in leukemic colony formation of AML (57-75% reduction) and HR-MDS BMMC-CD34+ samples (57-64%). These data are shown in **new Figure 3G**.

Further, we modified the results section to better indicate that cynomolgus monkeys are a relevant animal model for pharmacodynamic and toxicology assessment. AFM28 is species cross-reactivity to cynomolgus CD16, expressed by cynomolgus NK cells, and to cynomolgus CD123, expressed for instance by cynomolgus basophils, resulting in dose-dependent depletion of basophils in peripheral blood of cynomolgus monkeys as described in **Figure 7B**. No adverse clinical observations in cynomolgus were noted.

Altogether, we are confident that the NK cell-dependent mode of action of AFM28 has a better safety profile associated with more transient effects compared to conventional CAR-T cells. Indeed, CAR-T cells are more prone for long-lasting adverse effects due to their proliferative capacity and persistence, with the risk for restricting endogenous recovery from e.g. myelosuppression or on-target/off-tumor effects (Baroni et al., *J Immunother Cancer*. 2020). This is in line with preclinical data, suggesting that anti-CD123 CAR-NK cells in contrast to anti-CD123 CAR-T cells did not elicit endothelial injury and severe hematopoietic toxicity in a human endothelial murine model and in a humanized mouse model, respectively (Caruso, 2022, *Journal of Hematology & Oncology*). Moreover, early clinical data of a novel CD123-targeting CAR-T cell design including an on/off switch showed only transient myelosuppression that recovered upon treatment completion (Wermke et al., *Blood* (2021) 137 (22): 3145–3148). We also included new primary references on anti-CD123 CAR-T cells in the introduction and discussion.

Reviewer #2

Nature Communication paper review A-Summary of Key results: Jana-Julia Siegler et al. describe the preclinical profile of a novel bifunctional NK cell engager (NKCE) molecule targeting CD123 on AML blasts and CD16a on NK cells (AFM28), exhibiting cytotoxic activity against CD123-positive leukemic cells in vitro (with activity in CD64+ cells), against blasts from MDS & AML ex vivo (bone marrow patient samples in presence of healthy donor NK cells) and in vivo in a CD123+ AML mice model. Activity towards AML cell lines irrespective of their levels of CD123, CD64 expression or mutational status & activity towards leukemic stem cells are described. Direct inhibition of IL-3-dependent proliferation of AML cells is shown. PK/PD relationship and a favorable toxicity profile in non-human primates (NHP) are shown, which support clinical development. This molecule differs from other reported CD123-targeting NKCE by the antibody (Ab) format with a tetravalent bispecific CD123/CD16A innate cell engager, bivalent for each target.

B-Validity:

The manuscript is well written, and data clearly presented. C-Originality and significance: The specific CD123-CD16A NKCE represents a novel molecule although the design and therapeutic concept is similar to previously described NKCEs targeting other antigens (doi: 10.1080/19420862.2021.1950264). The generic NKCE platform was previously reported. The therapeutic concept is similar to previously described NKCEs targeting similar antigen (doi: 10.1038/s41587-022-01626-2). The identification of the cytotoxic activity of the NKCE against leukemic stem cells and progenitor cells in AML and MSD is novel.

D- Data & methodology:

Most of the experiments are well designed and data well-presented and robust, based on state-of-the-art methodologies. However, the ex vivo studies are performed using IL2-pretreated healthy donor NK cells as effector cells against patient primary AML cells: evaluating ex vivo activity using an autologous setting would represent experimental conditions more translatable to the clinical situation. A complementary in vivo design using adoptive transfer of human NK cells would complement the analysis of in vivo therapeutic efficacy.

We would like to thank the reviewer for this suggestion. We have generated two new data sets that suggest that AFM28 can effectively arm AML patients' endogenous NK cells for anti-leukemic activity to a comparable extent to healthy donors at least in some AML patients.

First, we show in an autologous setting using *ex vivo* cultures of prospectively-sampled fresh whole blood, derived from untreated newly diagnosed AML patients (n=6), that AFM28 could demonstrate activity against primary leukemic blasts. Responder samples (n=3) showed a mean lysis of 76% (range 59-90%), while non-responders (n=3) exhibited a mean lysis of 7% (range 0-22%). This novel data set is presented in the **new Figure 4A**. Responsiveness was further enhanced when allogeneic NK cells (used freshly or after *in vitro* expansion and cryopreserved) were added, increasing the proportion of responders (lysis >50%) to 100% (n=6/6) and overall increasing blast lysis to 87% (range 60-100%).

Second, we further explored the ADCC activity in a set of prospectively-sampled patient-derived NK cells, enriched from freshly drawn blood of AML patients (n=5), and tested their ADCC capacity against allogeneic primary leukemic blasts, compared to healthy NK (n=5) cells. Effective blast lysis was achieved by n=3 AML-derived and all n=5 healthy NK cell samples. This novel data set is presented in the **new Figure 4B**.

Moreover, we added a new *in vivo* data set on the adoptive transfer of human NK cells armed with AFM28 in irradiated fully immunodeficient hIL-15 NOG mice inoculated IV with EOL-1_Luc cells (**new Figure 6D-E** (previously Figure 5)). Tumor growth was markedly delayed when mice were treated with AFM28-armed NK cells, whereas AFM28 treatment alone failed to induce tumor control in this model due to the lack of responsive effector cells. Adoptive transfer of NK cells alone showed lower anti-tumor activity but only at early time points, suggesting rapid exhaustion of NK cells in the absence of AFM28.

E. Appropriate use of statistics and treatment of uncertainties.

The n of primary AML or MDS samples is low (n=5) and the n of the colony-forming Unit (CFU) assay very low (n=3 AML & n=3 MDS) to draw general conclusions given the heterogeneity of the disease. In Fig 5C, Kaplan-Meier plot shows survival upon treatment, with data represented as mean \pm SEM: the mean \pm SD for the median calculation seems more appropriate.

The number of experiments of the colony-forming unit (CFU) assays were increased to n=5 per indication (AML, MDS and healthy). We have now used at least n=5 primary patient samples per entity for all assays. We would like to emphasize again that these are rare and precious primary patient samples, which makes it difficult to obtain them in general and to obtain enough material to perform as many assays as possible with the same patient material.

The SEM in **Figure 6B** (previously Figure 5) and the **new Figure 6E** was replaced by the SD.

F-Conclusions:

the conclusions and data interpretation are valid and reliable

G-Suggested improvements:

The exact architecture of the NKCE format used should be clarified. It is described as a tetravalent bispecific CD123/CD16A innate cell engager. Does the construct contain an Fc domain? I suggest to add a scheme with relevant details as a supplementary figure.

AFM28 is a tetravalent bispecific humanized IgG1-scFv fusion antibody composed of two CD123 and two CD16A (Fc γ R1IIIA) binding domains (CD123/CD16A innate immune cell engager). AFM28 includes an Fc-silenced portion in the IgG1 backbone that is silenced for binding to Fc γ R while maintaining binding to the neonatal Fc receptor (FcRn) as was described in the **Supplementary Data**.

We added a scheme of AFM28 and its targeting domains and features to **Supplementary Figure 1**.

Thanks for adding the scheme that clarifies the structure & features of the AFM28 molecule on **Supp Fig1A**.

Regarding the activity towards leukemic stem cells (LSC), it seems to me important to investigate the potential activity of the compound on normal hematopoietic progenitor cell (HSPC) to see if the compound induces killing of LSC specifically, with normal progenitor capacity maintained (no or minimal HSPC killing?); the documentation of the differential CD123 expression level would be of interest.

We agree that evaluation of the normal healthy bone marrow compartment supports the assessment of the preclinical safety of AFM28. We added new *in vitro* experiments, assessing the colony-forming potential of healthy BMMC-derived CD34+ cells after treatment with AFM28 and allogeneic NK cells,

in addition to the already provided data on AML and HR-MDS. In these experiments, we could demonstrate that colony formation of healthy BMMC-CD34+ samples was only modestly affected after exposure to AFM28 (6-39%) compared to the robust reduction in leukemic colony formation of AML (57-75% reduction) and HR-MDS BMMC-CD34+ samples (57-64%). These data are shown in **new Figure 3G**.

While performing the additional CFU assays, we also determined CD123 expression on the enriched CD34+ cells from healthy donors. On CD34+CD38+ cells, the mean CD123 expression was 4.9% (range: 2.5-8.5%), most comparable to the CD45+CD34+CD38+ leukemic blasts, which were 71.8% (range: 51-84.2%) in AML and 50.1% (range: 30.5-62.4%) in MDS (see **Supplementary Figure 1 C-D**). On CD34+CD38- cells, mean CD123 expression was 4.9% (range: 2.5-8.5%), most comparable to CD45+CD34+CD38-CD117+ LSPCs, which was 71.5% (range: 60.3-91.4%) in AML and 63.4% (range: 33.9-80.1%) in MDS (see **Supplementary Figure 1 C-D**). These data support the assumption that CD123 expression appears to be lower on healthy HSPCs than on leukemic cells, which in turn explains why the number of colonies in the CFU assay in the healthy samples was less reduced by treatment with AFM28 than in the AML and MDS samples.

Thanks for adding the gating strategy of blasts and LSPCs, and the proportion of CD123+ cells of AML and HR-MDS patient-derived BMMC samples. I would not put these data in the same supp fig as the AFM8 structure but to a distinct one since this is not the same topic.

I don't see the CD123 expression on the CD34+ cells from healthy donors on Supp Fig1? It is I guess on Fig3G in main part of the manuscript? It would be useful to get the pictures of the CFU plates and a table with the CFU manual counts. Thanks

Evaluating *in vivo* an AML model with low CD123 antigen density would be a plus to confirm *in vitro* findings. The model used is not an AML & its mutational status is not described: an additional human AML or MDS tumor model with a defined mutational status would be better.

Regarding *in vivo* evaluation, the model used is not an AML & its mutational status is not described: an additional human AML tumor model with a defined mutational status or with low CD123 antigen density is missing, to confirm the potential of the molecule to deplete tumor cells irrespective of their mutational profile or CD123 expression level.

We agree with the reviewer that a more physiological approach would be more informative. We feel that the new data set using non-manipulated primary whole blood from different AML patients can provide the most compelling physiologically-relevant context. This novel data set is presented in the **new Figure 4A**. These new data show that AFM28 exposure can elicit anti-leukemic activity in an autologous setting in *ex vivo* cultures of prospectively-sampled fresh whole blood derived from untreated first diagnosis AML patients (n=6) with a variety of mutational profiles (see **Supplementary Table 3**). Responder samples (n=3) showed a mean lysis of 76% (range 59-90%), while non-responders (n=3) exhibited a mean lysis of 7% (range 0-22%). These results suggest the potential to arm the AML patient's endogenous NK cells for anti-leukemic activity.

The *in vivo* models, now further amended in the revised manuscript version in **Figure 6** (previously Figure 5), were used as an adequate model systems to demonstrate the proof-of-concept to address the two questions, (1) if AFM28 can induce tumor growth delay *in vivo* via endogenous innate immune components and (2) if AFM28 can enhance *in vivo* anti-tumor activity of adoptively-transferred human NK cells. The eosinophilic AML cell line EOL-1 was chosen in favor of an in-house established murine C1498 cell line with ectopic human CD123 overexpression, which showed CD123 expression even two to three orders of magnitude higher than detected on AML cells. We would like to state that the EOL-1 cell line faithfully belongs to the rare type of acute myeloid eosinophilic

leukemia, which has a KMT2A tandem duplication first described by Saito et al. Blood. Vol 66. No 6 1985: pp 1233-1240 (<https://www.dsmz.de/collection/catalogue/details/culture/ACC-386>). We have added the mutational status of the cell line to the **Supplementary Table 2**. We thank the reviewer for bringing this to our attention.

Thanks for adding these data addressing the different points raised. Regarding Fig 4, it is important to precise the E:T ratio in the autologous setting without addition of NK cells, or with addition of allogenic NK cells, in order to compare the cytotoxic activities measured at 24h, since the E:T ratio will impact the level of activity. Also the allogenic NK cells added are non expanded fresh ones or expanded frozen ones. Did you compare the phenotype of these different NK cells in term of basal activation? The authors should explain in the material and methods the protocol for expansion of NK cells and the phenotypes of the NK cells (hyperactivated?) post-expansion.

AFM28 treatment at 3 mg/kg alone failed to induce tumor control in the NSG mice model with adoptive transfer of human NK (activity only in mice treated with AFM28-armed NK cells). Did the authors try different amounts of human NK cells? Did they evaluate different NK cell batches given the variability between healthy donors? The material & method part is not detailed enough for this model.

The authors state that "Adoptive transfer of NK cells alone showed limited anti-leukemic activity and only at early time points in this model, suggesting rapid exhaustion of NK cells in the absence of AFM28": It is not clear if these observations were reproduced on at least 2 to 3 NK batches. If they is only one NK donor evaluated, no conclusion can be made.

In Fig. 6D, the increase of IL6 in animals treated with the compound and that develop ADA should be further documented.

We thank the reviewer for making us aware of this issue. We adjusted the order of statements in this section. In this respect, we added that the emerging ADA response in cynomolgus monkey resulted in reduced AFM28 exposure, which coincided with reduced PD activity and modest induction of IL-6 cytokine levels. Further, we also added that the emergence of the ADA response in cynomolgus monkeys, potentially a xenogeneic effect, is not predictive for ADA formation in humans (Swanson and Bussiere, Current Opinion in Microbiology 2012).

Thanks for separating the ADA+ or - NHPs.

Fig7B shows the activity of the compound in whole blood (WB) in term of basophil depletion and the CD16a expression on NK cells in WB: to interpret the data, it would be useful to know the NK:blast ratio in these samples and to correlate the observations of CD16a expression level modulations with the activity of the compound (basophil depletion)

In supplementary Figure 3B showing CD16a decrease at 0.1 µg/ml but an increase at 1 µg/ml: I would recommend to organize the data separating 3h & 24h since the E:T ratio of 1:1 does not give the same level of activity at 3h with this low E:T ratio and at 24h. It is important to read the modulation of CD16a expression at an active dose vs a poorly active one: the data of cytotoxic activity at these 2 time points are missing.

Also, given that the NKCE concept is already established, the manuscript is missing in mechanistic explorations of the advantage of targeting CD16A selectively as compared to several other CD16-based NKCE that can target CD16A and CD16B.

AFM28 was developed with a proprietary design to specifically redirect the anti-tumor activity of NK cells towards AML cells. Since NK cells only express the activating isoform CD16A and not CD16B, which may have rather an inhibitory role in neutrophils, AFM28 and other innate cell engagers specifically target CD16A. More relevantly in the context of AML, we believe that the benefit of specific CD16A targeting in contrast to pan FcγR targeting is illustrated by the experiments shown in **Figure 1**, demonstrating that a “non-Fc-specific” Fc-enhanced anti-CD123 IgG1 antibody fails to induce ADCC when AML cells express other FcγR. A similar observation has been reported for an NKp46-targeting NK cell engager (Gauthier et al., Nat. Biotech. 2023) referenced in this manuscript. Because of the bivalent CD16 targeting, specificity for CD16A is a hallmark of all innate cell engagers to prevent unwanted cross-linking of CD16A+ NK cells with other FcR+ cells including CD16B+ neutrophils and thus to prevent neutrophil killing. Another advantage of the innate cell engager design is the high-affinity bivalent CD16A targeting feature, enabling enhanced NK cell surface retention and reduced competition by serum IgG as compared to previous concepts that were based on lower-affinity monovalent CD16 targeting (Pinto et al., Trends Immunol. 2022).

Thanks for these explanations. Comparisons of CD16a specific targeting versus a FcγR targeting with a Fc moiety in vivo would be useful to confirm these statements.

8 AML & 6 MDS samples are described in the supplemental part but the activity data are shown only on 5 samples : could you explained why this is the case?

We thank the reviewer for bringing this uncertainty to our attention. This discrepancy is owing to the fact that we were not able to use the same set of patient samples for all assays due to the limited number and cell count of at least some of the primary patient samples. **Supplementary Tables 3-5** provide a precise breakdown of which sample was used for which assay. To avoid further confusion, we have added the following sentence to the methods section: “Details on the BM and PB donor characteristics and for which assay the samples were used are provided in **Supplementary Tables 3–5.**”

Thanks for these precisions.

H-References:

Relevant literature is appropriately cited.

I-Clarity and context:

abstract, introduction and conclusions are clearly written and appropriate. The paper is logically structured and well written. The CD123-CD16A NKCE show promising anti-tumor activity with activity on leukemic stem cells and a favorable safety profile. The conclusion that this reagent is ready for clinical testing is supported by the presented data. Inflammatory material: no inappropriate language

Reviewer #3 (Remarks to the Author):

This is a generally well written report on the pre-clinical development of the CD123 innate cell engager AFM28. The experiments are generally well designed and the methods and figures clearly delineated. As acknowledged by the authors CD123 is an attractive target for leukaemia-directed therapy with a view to directly eliminating LSC populations and/or interrupting IL3 trophic signalling. Despite its attractiveness, effective clinical targeting of CD123 with various agents has proven elusive. In the current paper Siegler and colleagues has described the next logical step in the development of CD123-directed therapy with the development of an NK-engaging tetra-valent molecule. The experiments that are outlined in the paper are a logical sequence of pre-clinical work up of a new drug but overall do not reveal any particularly new findings into the therapeutic application of this ICE.

The sensitivity of AML cells to allogeneic NK cells in the presence of AFM28 is a reasonable experimental design, but would have been more clinically relevant if performed using NK cells autologous to the AML targets. Presumably the samples containing leukaemic targets would also contain some residual NK populations. The confounding effect of HLA-disparity/non-self recognition of leukaemic targets by allogeneic NK cells limits the interpretation of this finding.

We would like to thank the reviewer for this suggestion. We provided a novel data set showing that AFM28 could demonstrate activity against primary leukemic blasts in an autologous setting in *ex vivo* cultures of prospectively-sampled fresh whole blood derived from untreated newly diagnosed AML patients (n=6). Responder samples (n=3) showed a mean lysis of 76% (range 59-90%), while non-responders (n=3) exhibited a mean lysis of 7% (range 0-22%). This novel data set is presented in the **new Figure 4A**. Responsiveness was further enhanced when allogeneic NK cells (used freshly or after *in vitro* expansion and cryopreserved) were added, increasing the proportion of responders (lysis >50%) to 100% (n=6/6) and overall increasing blast lysis to 87% (range 60-100%).

Importantly, the authors discuss the role of NK cell fitness in AML patients and its potentially limiting effect in ICE-based therapies. Whilst there is generally (sufficiently) preserved NK cell numbers and function in patients with newly diagnosed AML, further consideration and discussion should be undertaken regarding how this may change following chemotherapy delivery and/or in the setting of relapsed disease, which is likely to reflect the population in which these novel agents are to be initially trialed.

In addition to **new Figure 4A**, we further explored the ADCC activity of patient-derived NK cells, enriched from freshly drawn blood of AML patients (n=5) and tested against allogeneic primary leukemic blasts of the same AML patient, compared to healthy NK (n=5) cells. This novel data set is presented in the **new Figure 4B**. Effective blast lysis was achieved by n=3 AML-derived and all n=5 healthy NK cell samples. Collectively, these results suggest that AFM28 can effectively arm AML patients' endogenous NK cells for anti-leukemic activity to a comparable extent to healthy donors at least in some AML patients.

Moreover, we added available primary references to the discussion, describing the status of NK cells of AML patients after relapse, after chemotherapy and after complete remission in the context of the potential for AFM28 efficacy as a single agent or in combination with allogeneic NK cell therapy.

Further consideration and discussion should be undertaken as to the expression of inhibitory NK receptor ligands on LSCs targets, which may limit NK cell activation *in vivo*.

The authors would like to thank the reviewer for this recommendation. Generally, there is only few knowledge on the expression of NK cell inhibitory ligands as well as NK cell activating ligands. Most prominently it has been shown that LSC-like cells express reduced levels of NKG2D ligands, which has been suggested to support immune evasion from NK cells (Paczulla et al., Nature 2019). Further these authors reported high frequencies of AML cells positive for the NK cell activating ligands CD112 and CD155, whereas low frequencies were positive for B7-H6 and PD-L1. Moreover, immunomodulating molecules such as the NK cell receptors CD44, CD96 and TIM-3 have been described to be expressed on AML cells including LSC-like cells (Inagaki Y et al., Cell Stem Cell, 2010; Stelmach and Trumpp, Haematologica 2023).

We amended the discussion as follows: "It remains to be investigated whether additional targeting of activating and inhibitory NK cell ligands expressed by leukemic blasts including LSC-like cells (Stelmach and Trumpp, Haematologica 2023) enhances ADCC to further restrict the risk for minimal residual disease."